# Genome-wide single-molecule analysis of long-read DNA methylation reveals heterogeneous patterns at heterochromatin that reflect nucleosome organisation

Lyndsay Kerr[1], Ioannis Kafetzopoulos[2,3], Ramon Grima[4], Duncan Sproul[2]*

**1** MRC Institute of Genetics and Cancer, University of Edinburgh, Edinburgh, United Kingdom, **2** MRC Human Genetics Unit and CRUK Edinburgh Centre, Institute of Genetics and Cancer, University of Edinburgh, Edinburgh, United Kingdom, **3** Current address: Altos Labs Cambridge Institute, Cambridge, United Kingdom, **4** School of Biological Sciences, University of Edinburgh, Edinburgh, United Kingdom

* d.sproul@ed.ac.uk

**Data Availability Statement:** All data used in this study have been generated and made publicly available by others. GM24385 Nanopore

## Abstract

High-throughput sequencing technology is central to our current understanding of the human methylome. The vast majority of studies use chemical conversion to analyse bulk-level patterns of DNA methylation across the genome from a population of cells. While this technology has been used to probe single-molecule methylation patterns, such analyses are limited to short reads of a few hundred basepairs. DNA methylation can also be directly detected using Nanopore sequencing which can generate reads measuring megabases in length. However, thus far these analyses have largely focused on bulk-level assessment of DNA methylation. Here, we analyse DNA methylation in single Nanopore reads from human lymphoblastoid cells, to show that bulk-level metrics underestimate large-scale heterogeneity in the methylome. We use the correlation in methylation state between neighbouring sites to quantify single-molecule heterogeneity and find that heterogeneity varies significantly across the human genome, with some regions having heterogeneous methylation patterns at the single-molecule level and others possessing more homogeneous methylation patterns. By comparing the genomic distribution of the correlation to epigenomic annotations, we find that the greatest heterogeneity in single-molecule patterns is observed within heterochromatic partially methylated domains (PMDs). In contrast, reads originating from euchromatic regions and gene bodies have more ordered DNA methylation patterns. By analysing the patterns of single molecules in more detail, we show the existence of a nucleosome-scale periodicity in DNA methylation that accounts for some of the heterogeneity we uncover in long single-molecule DNA methylation patterns. We find that this periodic structure is partially masked in bulk data and correlates with DNA accessibility as measured by nanoNOMe-seq, suggesting that it could be generated by nucleosomes. Our findings demonstrate the power of single-molecule analysis of long-read data to understand the structure of the human methylome.

sequencing data are described https://labs.epi2me.io/gm24385_2020.09/ and raw data are available from the Amazon bucket s3://ont-open-data/gm24385_2020_09/. Nanopore sequencing data from GM12878 cells were downloaded from the Amazon bucket s3://nanopore-human-wgs/rel6/MultiFast5Tars/ and are also available from the European Nucleotide Archive (ENA) accession PRJEB23027 (https://www.ebi.ac.uk/ena/browser/view/PRJEB23027). Raw GM12878 nanoNOMe-seq data are available from the NCBI Sequence Read Archive (SRA) accession SRP173931 (https://trace.ncbi.nlm.nih.gov/Traces/index.html?view=study&acc=SRP173931).

**Funding:** LK is a cross-disciplinary post-doctoral fellow supported by funding from the University of Edinburgh and Medical Research Council (MC_UU_00009/2). IK is funded by a CRUK PhD studentship (C157/A25186) and The A. G. Leventis Foundation Educational Grant (18736). RG is supported by Leverhulme Trust research awards (RPG-2018-423 and RPG-2020-327). DS is a Cancer Research UK Career Development fellow (reference C47648/A20837). LK, IK and DS all received salaries from their respective funders. The funders had no role in study design, data collection and analysis, decision to publish, or preparation of the manuscript.

**Competing interests:** The authors have declared that no competing interests exist.

## Author summary

DNA methylation is an epigenetic DNA modification that is often associated with the repression of gene expression. Correct patterning of DNA methylation is crucial in development and for normal cellular function. Aberrant patterns of DNA methylation are also observed in diseases such as cancer. Here, we examine DNA methylation patterns within long DNA molecules to identify how methylation heterogeneity varies throughout the human genome within single molecules. We find that single molecule DNA methylation heterogeneity varies widely, with some genomic regions, particularly those away from genes, having very heterogeneous patterns. In contrast, we find that the regions of the genome around active genes have more homogeneous, uniform methylation states. When we examined the nature of heterogeneous methylation patterns, we find they have an underlying periodicity. We also find that this periodicity is reflective of the single molecule placement of nucleosomes, proteins that are important in the packaging of DNA. Overall, our study indicates that DNA methylation heterogeneity shows wide variation across the human genome and that DNA packaging could play a role in shaping DNA methylation patterns at the level of single molecules.

## Introduction

DNA methylation is an epigenetic mark commonly associated with gene repression [1]. In mammals it is prevalent on cytosines within CpG dinucleotides, with 70–80% of CpGs being methylated in most cell types [2].

High-throughput sequencing approaches have resulted in the description of many human methylomes using bisulfite conversion in combination with whole-genome sequencing (Whole-Genome Bisulfite Sequencing, WGBS) [3, 4]. Analysis of WGBS data has confirmed the pervasive nature of CpG methylation in human genomes and uncovered differences in the DNA methylation patterns between different cell types and tissues [5, 6]. They have also detailed differences in DNA methylation occurring in human disease, particularly cancer [7].

These insights have arisen from the analysis of bulk DNA methylation patterns. High-throughput bisulfite sequencing data can also be analysed at the single-read level to understand single-molecule heterogeneity in DNA methylation patterns. Analyses of single-read patterns have been used to quantify the degree of inter-molecular methylation heterogeneity in human samples [8, 9]. Further analyses have quantified differences in this heterogeneity between normal and cancer cells using information theory [10]. DNA methylation heterogeneity has been shown to change over time in cell culture [11] and in acute myeloid leukemia upon relapse [12]. Inter-molecular methylation heterogeneity has been attributed to epi-polymorphisms [8] and allele-specific methylation [13]. The mixture of cell types in a sample is also an important source of DNA methylation heterogeneity [14, 15], that could explain part of the inter-molecular heterogeneity seen in these analyses.

However, the degradation of DNA caused by bisulfite conversion [16] and the widespread use of Illumina sequencing technology limits single-molecule analyses of WGBS data to short regions of a few hundred basepairs. The limited number of CpGs captured in these reads restricts analyses of intra-molecular DNA methylation heterogeneity. Nanopore sequencing enables the generation of reads over a megabase in length [17], four orders of magnitude longer than those assayed by WGBS. DNA modifications including 5-methylcytosine can also be directly detected in these reads without chemical conversion [18]. This potentially enables the

analysis of molecular heterogeneity in DNA methylation patterns at a far greater scale than previously examined.

Here we conduct a genome-wide analysis of single-molecule DNA methylation patterns in long reads derived from Nanopore sequencing in order to understand the nature of large-scale intra-molecular DNA methylation heterogeneity in the human genome. Our work demonstrates that intra-molecular DNA methylation heterogeneity is abundant in heterochromatin, which possesses oscillatory methylation patterns that are partially masked in bulk data and are likely to be generated by nucleosomes.

## Results

### Large-scale methylation heterogeneity is underestimated by bulk analysis of methylomes

To understand the nature of large-scale heterogeneity in DNA methylation patterns, we analysed Nanopore sequencing data derived from GM24385 lymphoblastoid cells and released by Oxford Nanopore Technologies. This consisted of 6, 289, 480 reads aligned to autosomes with a mean length of 24.6kb. The mean coverage for autosomal reference CpGs was 39.1. We validated key results using Nanopore GM12878 data, [19] (see Methods) which consisted of 10, 784, 132 reads aligned to autosomes with a mean length of 8.7kb and with mean autosomal reference CpG coverage of 18.6. We used the GM24385 data for primary analysis due to its longer mean read length and its higher coverage.

We first asked how large-scale intra-molecular DNA methylation heterogeneity varied across single reads. We therefore calculated the mean level of methylation for all aligned single GM24385 reads along with the coefficient of variation and correlation between neighbouring CpGs to capture intra-molecular variation (Fig 1a). To focus our attention on large-scale DNA methylation patterns, and reduce noise associated with calculating statistics from a low number of CpGs, only aligned reads with methylation called for $\geq$100 CpG sites were considered (2, 908, 181). All of these single-read measurements varied considerably across aligned reads suggesting that substantial large-scale intra-molecular DNA methylation heterogeneity exists in human cells (Fig 1b–1d).

We then asked how these single-read measurements compared to bulk-level measurements (Fig 1a). To facilitate this comparison, we segmented the autosomal genome into 100kb non-overlapping windows and calculated the mean of each of the single-read statistics for each window using the GM24385 reads that aligned entirely within that windows (total of 1, 707, 118 reads for all windows). We compared these mean single-molecule measures to bulk-level statistics for the same windows. We calculated the bulk statistical measure for a window by first calculating the mean methylation level of each CpG in the window using all GM24385 reads in which it had a defined methylation state. We then calculated the bulk statistics using these mean CpG methylation values. We hypothesised that the bulk and single-read approaches would provide similar estimates of mean methylation levels but differ in their quantification of variation. Measurements of the mean single-read methylation levels in the genomic windows were significantly correlated with bulk mean levels of DNA methylation within the same windows (Fig 1e, Spearman correlation = 0.943, $p < 2.20 \times 10^{-16}$). While a two-sided paired t-test indicated significant difference in the bulk means compared to the single-read means, the absolute relative difference of the single-read means compared to the bulk means was small (mean absolute relative difference of 0.0375). As the single-read and bulk estimates of mean methylation are mathematically equivalent, these small differences are likely to be caused by slightly different read and CpG exclusion criteria between the two approaches. As expected,

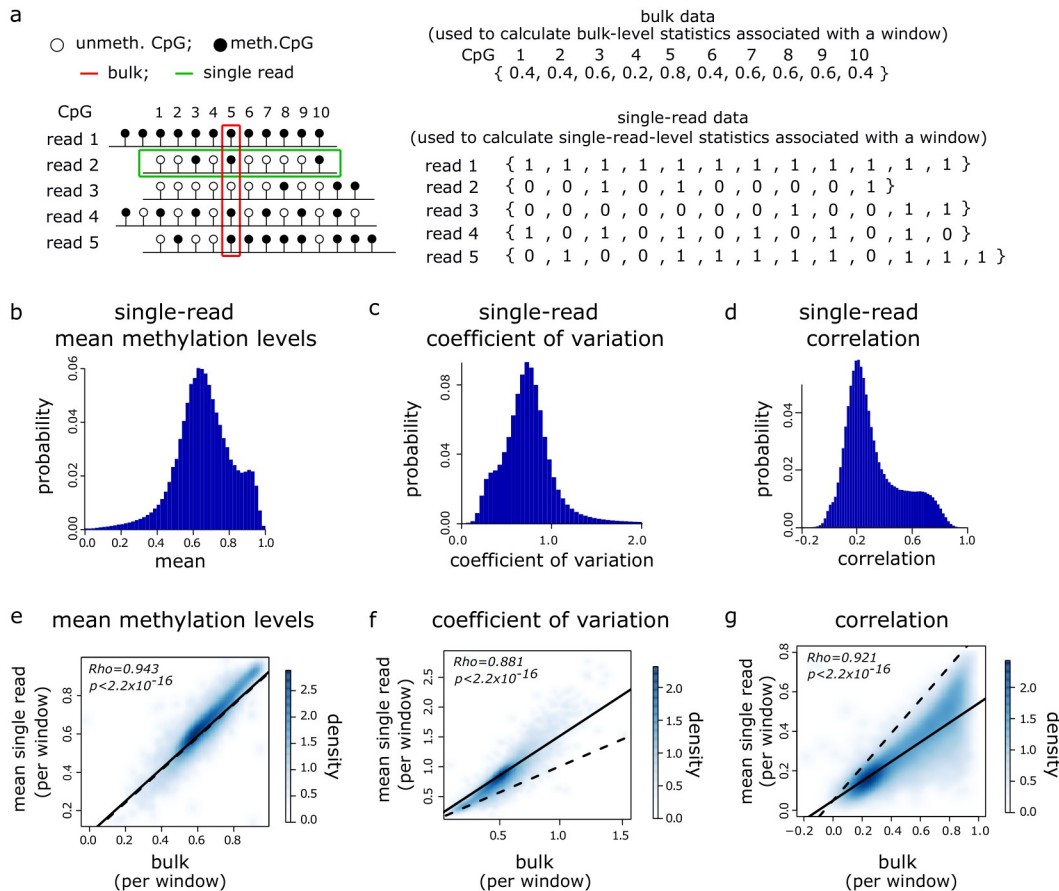

**Fig 1. Large-scale methylation heterogeneity is underestimated by bulk analysis of methylomes. a** Left: schematic of analysis approach. Example single-read (green box) and bulk (red box) analysis for 5 hypothetical reads overlapping 10 common CpGs are shown. Right: the bulk methylation level of a CpG is its number of methylated observations divided by its total number of unmethylated and methylated observations. The red box highlights information used to calculate the bulk methylation level of CpG 5. Single-read data is obtained by creating a vector, $v$, for each read, where $v_i = 0, 1$ if CpG $i$ in the read is unmethylated or methylated respectively. The green box highlights information used to construct $v$ for read 2. **b–d** Histograms showing the distributions of single-read statistics for the 2, 908, 181 GM24385 reads with methylation states for ≥100 CpGs. (**b**) mean methylation levels, (**c**) coefficient of variation and (**d**) correlations between neighbouring CpGs. **e–g** Density scatter plots of GM24385 mean single-read vs. bulk statistics in 100kb genomic windows ($n = 26, 687$). (**e**) mean methylation levels, (**f**) coefficient of variation and (**g**) correlations between neighbouring CpGs. Dashed and solid lines show lines of identity and linear models fitted to the data respectively. Spearman correlations are shown along with p-values from paired t-tests comparing bulk to mean single-read statistics.

these results suggest that single-read measurements capture largely the same information regarding overall mean methylation levels as bulk measurements.

We next considered how single-read measurements of heterogeneity compared to their bulk counterparts. Like mean methylation levels, the mean single-read and bulk measurements of the coefficient of variation and correlation were significantly correlated (Fig 1f and 1g, Spearman correlation = 0.881 and 0.921, respectively, $p < 2.20 \times 10^{-16}$ in both cases). However, mean single-read coefficients of variation observed in the genomic windows were significantly greater than that of the equivalent bulk measurements (Fig 1f, paired t-test $p < 2.20 \times 10^{-16}$, mean absolute relative difference of 0.338) and a linear model fitted to the two deviated from the line of identity (Fig 1f solid vs. dashed line). We also observed that mean single-read correlations were significantly lower than that of the bulk measurements

(Fig 1g, paired t-test p $< 2.20 \times 10^{-16}$, mean absolute relative difference of 0.621) and a linear model fitted to the two deviated from the line of identity (Fig 1g solid vs. dashed line). To ensure that these results were not specific to the window size used, we repeated the analysis using 50kb and 200kb genomic windows observing similar results as when 100kb windows were used (S1 Fig).

These comparisons of single-molecule to bulk measurements suggest that the analysis of DNA methylation in single Nanopore reads captures large-scale intra-molecular DNA methylation heterogeneity that is missed by analysis of bulk DNA methylation patterns.

## Intra-molecular heterogeneity varies across the human genome

Having established that significant large-scale intra-molecular DNA methylation heterogeneity exists in the human methylome, we next asked how this heterogeneity was organised across the genome.

Many measures previously used to quantify single-molecule methylation heterogeneity from short-read data [20] are difficult to apply to long reads because the number of possible states increases with the number of CpGs in the read. We therefore measured single-read methylation heterogeneity using the correlation in methylation state between nearest-neighbour CpGs (see Eq (4)). We chose to measure single-molecule heterogeneity using the correlation rather than the coefficient of variation since the latter is dependent on the overall mean methylation level and does not take into account information relating to the methylation states of neighbouring CpGs. A high correlation indicates high agreement in the methylation state of nearest-neighbour CpGs and thus low intra-molecular heterogeneity. On the other hand, a low correlation indicates high disagreement in the methylation state of nearest-neighbour CpGs and thus high intra-molecular heterogeneity.

To ask how intra-molecular heterogeneity varied across the genome, we calculated the correlation associated with each of the 2, 908, 181 GM24385 reads containing methylation information for $\geq 100$ CpGs that mapped to autosomes. The resulting distribution of individual read correlations was weakly bimodal (Fig 2a). We observed a significantly different unimodal distribution (Kolmogorov-Smirnov test, $p < 2.20 \times 10^{-16}$) characterised by lower correlation values when we calculated the correlation using reads simulated to have a random intra-molecular methylation pattern (Fig 2a, see Methods). These results suggest that reads had methylation patterns that were non-random and that intra-molecular methylation heterogeneity might vary by location in the genome.

To understand whether the degree of methylation heterogeneity varied by location in the genome, we examined the mean correlation across GM24385 reads entirely aligned within 100kb windows (Fig 2b, [21]). We then compared the distribution of mean correlation to that obtained when we assigned random reads to each genomic window (shuffled correlation). The distribution of mean correlation for genomic windows was bimodal and significantly different from the unimodal distribution observed for the shuffled correlation (Kolmogorov-Smirnov test, $p < 2.20 \times 10^{-16}$, Fig 2c and 2d). Moreover the peak of the unimodal shuffled correlation distribution lies between the two peaks associated with the bimodal mean correlation distribution. This is consistent with some regions of the human genome possessing a lower degree of intra-molecular heterogeneity than would be expected by chance, and other regions displaying a higher degree of intra-molecular heterogeneity. To ensure that this variation in the degree of methylation heterogeneity across the genome was not specific to GM24385 cells, we also examined single-read correlations for the 1, 327, 908 GM12878 reads which had at least 100 CpGs.

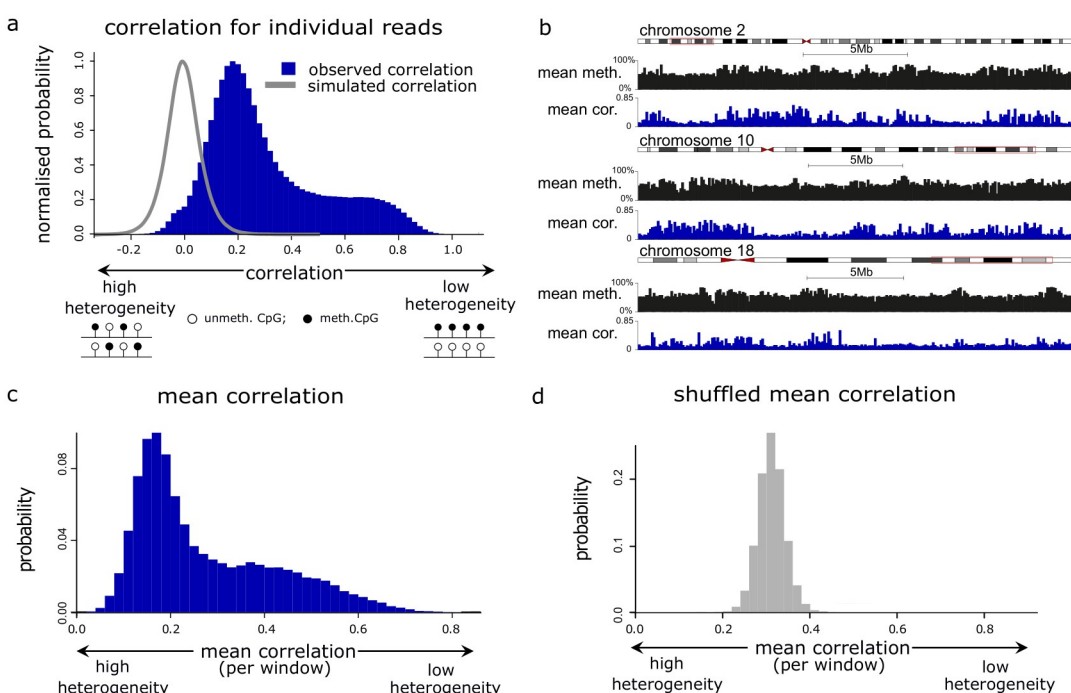

**Fig 2. Intra-molecular heterogeneity varies across the human genome. a** Histogram showing the correlation distribution for the 2, 908, 181 GM24385 reads containing methylation states for $\geq$ 100 CpGs. This is compared to the correlation calculated from reads simulated to have a random intra-molecular methylation pattern (grey line). **b** Genome browser plots showing the mean methylation levels and mean correlation in 100kb windows for selected genomic windows in GM24385. **c** Histogram of the mean correlation distribution for 100kb genomic windows in GM24385 ($n$ = 26, 951), calculated using reads that align entirely within each window. **d** Histogram showing the mean shuffled correlation in 100kb windows ($n$ = 26, 951), calculated using random GM24385 reads rather than those that align to each window.

Both the distributions of read correlations and mean window correlations for GM12878 cells were bimodal (S2a and S2b Fig).

To validate that intra-molecular methylation heterogeneity varies across the genome, we then conducted a similar analysis using an alternative measure of intra-molecular heterogeneity, the Read Transition Score (RTS; [22], Eq 11, S2c Fig). The RTS measures the probability that two nearest-neighbour CpGs have different methylation states and ranges from 0 to 1. An RTS of 1 indicates that no nearest-neighbour CpGs have the same methylation state and an RTS of 0 indicates that only a single methylation state is observed in the read (S2c Fig). Hence we take a low RTS to be indicative of low heterogeneity within the single-molecule pattern and a high RTS to be indicative of high heterogeneity within the single-molecule pattern. The RTS for individual reads was again observed to be bimodally distributed (S2d Fig) and showed significant negative correlation with the single-read correlation (Spearman correlations = −0.666, $p < 2.20 \times 10^{-16}$, S2e Fig). Similarly, the mean RTS in 100kb windows showed significant negative correlation with the mean correlation in 100kb windows (Spearman correlations = −0.768, $p < 2.20 \times 10^{-16}$, S2f Fig). In addition, the distribution of mean RTS observed in 100kb windows was significantly different from that obtained from shuffled reads, consistent with the degree of intra-molecular heterogeneity varying across the human genome (S2g and S2h Fig, Kolmogorov-Smirnov test, $p < 2.20 \times 10^{-16}$).

Taken together, these analyses suggest that the degree of large-scale intra-molecular DNA methylation heterogeneity varies between different regions of the human genome.

## Intra-molecular and inter-molecular methylation heterogeneity co-occur in the human genome

Intra-molecular methylation heterogeneity can occur in a manner that is consistent or inconsistent across reads arising from the same region of the genome. This means that a low correlation in the methylation state between nearest-neighbour CpGs could be compatible with either high or low inter-molecular methylation heterogeneity (S3a Fig).

To understand the degree of inter-molecular heterogeneity associated with regions of the genome displaying high intra-molecular methylation heterogeneity, we computed the measure $d_i$. This quantifies the degree of inter-molecular heterogeneity across reads for a given CpG as $d_i = |m_i - 0.5|$ for a CpG $i$ with methylation level $m_i$. This ranges from 0, which indicates maximum inter-read heterogeneity when the unmethylated and methylated states are observed equally across reads, to 0.5, when a CpG has a uniformly methylated or unmethylated state.

We calculated $d_i$ for every CpG in the genome using the GM24385 data before comparing the values observed for regions of the genome associated with the highest or lowest degree of intra-read methylation heterogeneity as measured by the correlation. The distribution of $d_i$ values for the 10% of 100kb genomic windows with the highest mean correlation were significantly skewed towards high values of $d_i$ in comparison to the background of all genomic windows (Fig 3a and 3b, $p < 2.20 \times 10^{-16}$, Kolmogorov-Smirnov test). This indicates that these regions display low inter-read heterogeneity as well as low intra-read heterogeneity. In contrast the $d_i$ scores of CpGs found in the 10% of 100kb genomic windows with the lowest mean correlation were significantly lower than the background of all genomic windows (Fig 3a and 3b, $p < 2.20 \times 10^{-16}$, Kolmogorov-Smirnov test) suggesting that they also show a greater degree of inter-read heterogeneity. In order to examine how inter-read heterogeneity and intra-read heterogeneity correlated genome-wide, we then examined the association between mean $d_i$ value and mean correlation in all 100kb genomic windows. In agreement with high intra-molecular heterogeneity coinciding with high inter-molecular heterogeneity, we observed a significant positive correlation between mean $d_i$ and mean correlation (S3b Fig, Spearman correlation = 0.632, $p < 2.20 \times 10^{-16}$).

These analyses of CpG methylation levels demonstrate that regions of the genome with a high degree of intra-molecular DNA methylation heterogeneity also posses high inter-molecular methylation heterogeneity.

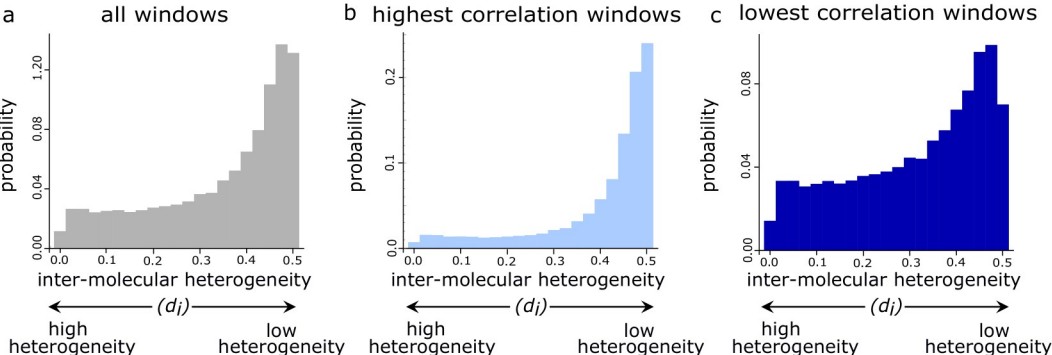

**Fig 3. Intra-molecular and inter-molecular methylation heterogeneity co-occur in the human genome. a–c** Histograms showing the distributions of $d_i$ in GM24385 **a** for the 26, 205, 980 CpGs in all windows, **b** for the 4, 504, 144 CpGs in the 10% of windows with the highest correlation and **c** for the 1, 535, 028 CpGs in the 10% of windows with the lowest correlation.

## Heterochromatin possesses heterogeneous methylation patterns

To understand the nature of the regions of the human genome which had the greatest degree of intra-molecular DNA methylation heterogeneity, we asked whether they were associated with particular genomic annotations or chromatin states.

Using the GM24385 data we first analysed the 10% of 100kb genomic windows with the lowest mean correlation (i.e. the 10% most heterogeneous windows). We observed that these genomic windows were significantly depleted in CpG islands compared to the background of all genomic windows (fold-enrichment of 0.0154, Wilcoxon test, $p < 2.20 \times 10^{-16}$) but showed no significant depletion or enrichment in genes (fold-enrichment of 0.936, Wilcoxon test, $p = 0.566$). The least heterogeneous genomic windows (the 10% with the highest mean correlation), were significantly enriched in both genes and CpG islands (fold-enrichments of 1.21 and 4.18, Wilcoxon tests $p = 1.09 \times 10^{-6}$ and $p < 2.20 \times 10^{-16}$, respectively).

To further understand the characteristics of regions displaying high methylation heterogeneity, we cross-referenced them to chromatin state data generated by the ENCODE project [23, 24]. These projects have used hidden Markov models to partition the genome into distinct chromatin states (chromHMM). Consistent with their observed enrichment in genes and CpG islands, the least heterogeneous genomic windows were significantly enriched in chromHMM-defined promoters, enhancers and transcribed regions from GM12878 lymphoblastoid cells (Fig 4a, Wilcoxon tests with $p < 2.20 \times 10^{-16}$ in each case). They were also significantly enriched in insulators and polycomb-repressed regions (Fig 4a, Wilcoxon tests, $p = 6.03 \times 10^{-12}$ and $p < 2.20 \times 10^{-16}$, respectively). In contrast, the most heterogeneous windows were significantly enriched in chromHMM-defined heterochromatic regions (Fig 4b, Wilcoxon test with $p < 2.20 \times 10^{-16}$). We observed similar enrichment results when the analysis was repeated using 50kb and 200kb genomic windows (S4a and S4b Fig) suggesting that our observations were independent of the choice of genomic window size. To ensure that these chromHMM enrichment results were not specific to GM24385 cells, we repeated the analysis for the least and most heterogeneous 100kb windows identified in GM12878 cells and observed analogous significant enrichments to our analysis of GM24385 cells (S4c Fig).

In some somatic cells, heterochromatin is associated with regions of intermediate methylation levels that are termed partially methylated domains (PMDs) [3, 25]. To further assess the correspondence between heterochromatin and intra-molecular DNA methylation heterogeneity, we therefore called 678 PMDs in GM24385 cells from our Nanopore DNA methylation data (S4d Fig) and defined the rest of the genome as 582 non-PMDs. GM24385 PMDs showed substantial overlap with those previously defined in 11 other lymphoblastoid cells from WGBS data (Jaccard = 0.804) [26]. Consistent with their enrichment in the chromHMM-defined heterochromatin state, genomic windows with the lowest mean correlation were significantly enriched in GM24385 PMDs (Fig 4c, Wilcoxon test $p < 2.20 \times 10^{-16}$) with 60.0% of these windows overlapping PMDs by at least half the window length. Those windows with the highest mean correlation were significantly enriched in non-PMDs (Fig 4c, Wilcoxon test $p < 2.20 \times 10^{-16}$) with 97.6% of these windows overlapping non-PMDs by at least half the window length. We observed similar enrichment results when the analysis was repeated using 50kb and 200kb genomic windows (S4e Fig) suggesting that overlap between intra-molecular methylation heterogeneity and PMDs was independent of the choice of genomic window size. To ensure that these PMD/non-PMD enrichment results were not specific to GM24385 cells, we also called 1,140 PMDs in GM12878 cells (designating the rest of the genome as 968 non-PMDs). We then repeated the analysis for the least and most heterogeneous windows in GM12878 cells finding the most heterogeneous windows to be significantly enriched in PMDs and the least heterogeneous windows to be significantly enriched in non-PMDs (S4f Fig).

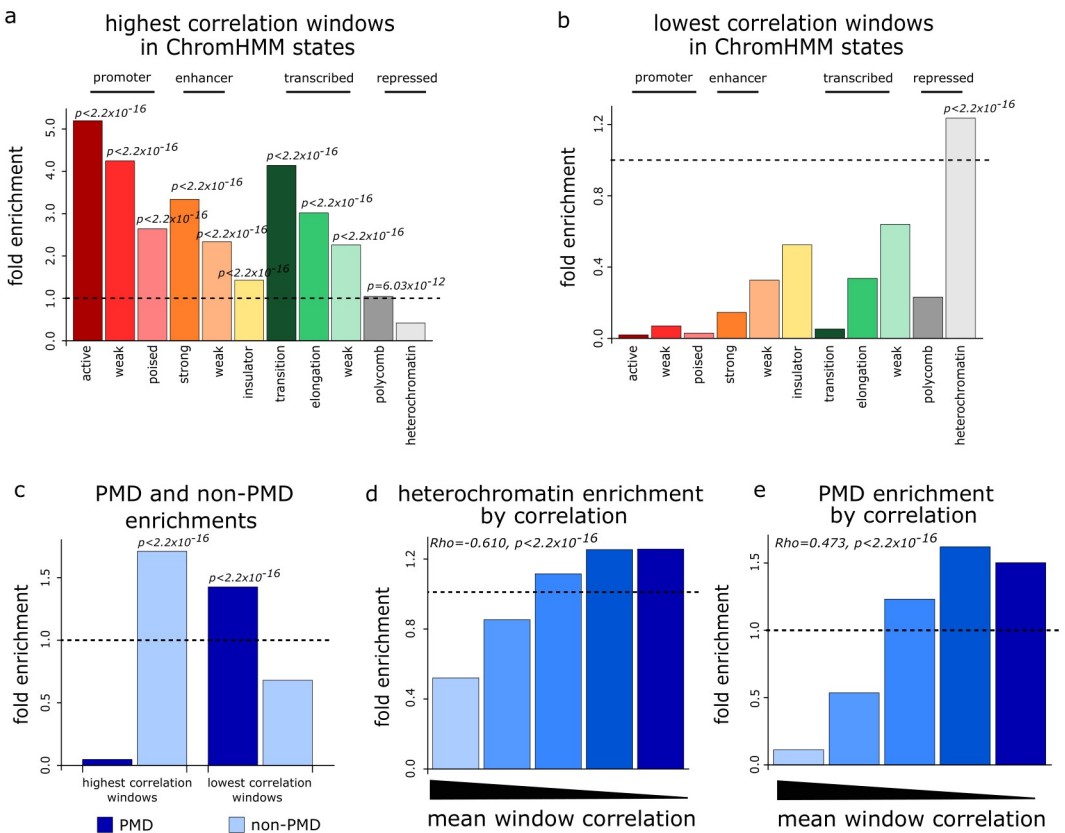

**Fig 4. Heterochromatin possesses heterogeneous methylation patterns. a,b** Barplots showing enrichment of the 10% least (**a**) and most (**b**) heterogeneous 100kb windows in chromHMM states defined in GM12878 lymphoblastoid cells. **c** Barplot showing the enrichment of the 10% least and most heterogeneous 100kb windows in PMDs and non-PMDs. **d** Barplot showing enrichment of continuous heterogeneity groups of genomic windows (defined from the ranked mean correlation) in the chromHMM heterochromatin annotation from GM12878 lymphoblastoid cells. **e** Barplot showing enrichment of continuous heterogeneity groups of genomic windows (defined from the ranked mean correlation) in PMDs. In **d, e** the Spearman correlations (Rho) between mean window correlation and window overlap with heterochromatin or PMDs, respectively, are also shown along with the relevant p-values.

To examine how methylation heterogeneity associates with heterochromatin and PMDs more generally, we calculated the correlation between the mean correlation in 100kb genomic windows and the proportion of window overlapping the chromHMM heterochromatin state or PMDs. Since methylation heterogeneity increases as correlation decreases, a negative correlation between correlation and overlap with a genomic annotation is indicative of the annotation being associated with high heterogeneity. Mean window correlation showed a significant negative correlation with both the proportion of the window covered by the chromHMM heterochromatin state and the proportion of the window covered by PMDs (Fig 4d and 4e, Spearman correlation = −0.610 and −0.473 respectively, both $p < 2.20 \times 10^{-16}$). At the single-molecule level, we also found significant negative correlations between a read's correlation and the proportion of the read that overlapped the heterochromatic state and PMDs (Spearman correlation = −0.474 and −0.351 respectively, both $p < 2.20 \times 10^{-16}$) confirming that the association between intra-molecular DNA methylation heterogeneity and heterochromatin was maintained at the single-read level.

Taken together, these analyses suggest that the regions of the human genome displaying the greatest degree of large-scale intra-molecular methylation heterogeneity are heterochromatic partially methylated domains.

## Periodic DNA methylation patterns are observed in single reads with high heterogeneity

Having shown that heterogeneity in DNA methylation patterns is observed in heterochromatin, we now sought to determine the nature of intra-molecular DNA methylation heterogeneity in these regions in order to understand its potential causes.

In order to do so, we first defined groups of single GM24385 reads exhibiting low or high intra-molecular methylation heterogeneity by fitting the sum of two Gaussian distributions to the bimodal distribution of correlations (Fig 2a: low correlation reads, i.e. high heterogeneity reads, and high correlation reads, i.e. low heterogeneity reads, correspond to the left and right peaks respectively). High heterogeneity reads were significantly enriched in heterochromatin and PMDs, while low heterogeneity reads were enriched in promoters, enhancers, insulators, transcribed regions, polycomb-repressed regions and non-PMDs, confirming the associations observed in previous analyses (Wilcoxon tests, $p < 2.20 \times 10^{-16}$ in each case).

We now sought to determine how the correlation between CpGs depends on the distance between them. Analysis of bulk WGBS data has reported oscillatory distance-dependent correlations in mean methylation levels within PMDs [27–29]. Given that we observed high intra-molecular heterogeneity associated with PMDs, we wondered whether oscillatory DNA methylation patterns might be present at the single-molecule level and account for some of the heterogeneity we measure. We therefore calculated the correlation in methylation state for pairs of CpGs separated by different distances in single GM24385 reads. Calculating these correlations for all possible distances in each individual long read is prohibitively computationally expensive. To reduce the computational burden, we therefore calculated distance-dependent correlations within the low and high heterogeneity GM24385 reads for distances up to a maximum of 500bp. As expected, mean distance-dependent correlations were higher for low heterogeneity reads than high heterogeneity reads across distances of 2–500bp (Fig 5a). Within both low and high heterogeneity reads, the mean correlation was highest for CpGs located close to each other and reduced as the distance between CpGs increased (Fig 5a). However, after initially falling, the mean distance-dependent correlation for high heterogeneity reads then increased, showing oscillatory behaviour (Fig 5a). No clear oscillatory behaviour was apparent in the mean distance-dependent correlation profile for low heterogeneity reads (Fig 5a). To ensure that this periodicity was not a feature specific to GM24385 cells, we repeated the analysis using the data from GM12878 cells. GM12878 high heterogeneity reads displayed oscillations that were similar to those seen in GM24385 cells and that were largely missing in low heterogeneity GM12878 reads (S5a Fig).

To determine whether the oscillatory behaviour within high heterogeneity reads could be seen beyond distances of 500bp, we then computed distance-dependent correlations up to 2kb between CpGs within the high heterogeneity GM24385 reads. While the oscillations decrease in magnitude with distance (Fig 5b), we observed that they continued up to 2kb within high heterogeneity reads (Fig 5c). In order to determine the wavelength of these oscillations, we examined the power spectrum associated with the distance-dependent correlation up to 2kb for the high-heterogeneity reads. The frequency at which this power spectrum peaked was 0.0335 corresponding to a wavelength of 187.6bp with the power being at least half of the maximum value at the peak for frequencies within the range 0.0313–0.0357 (S5b and S5c Fig). The spectral width therefore indicates that the oscillations have wavelength in the approximate range 176.0–200.7bp. Since we previously found PMDs to be associated with highly heterogeneous single-molecule methylation patterns, we asked whether oscillatory patterns were also visible in reads aligning to PMDs. Indeed, the distance-dependent correlations calculated for

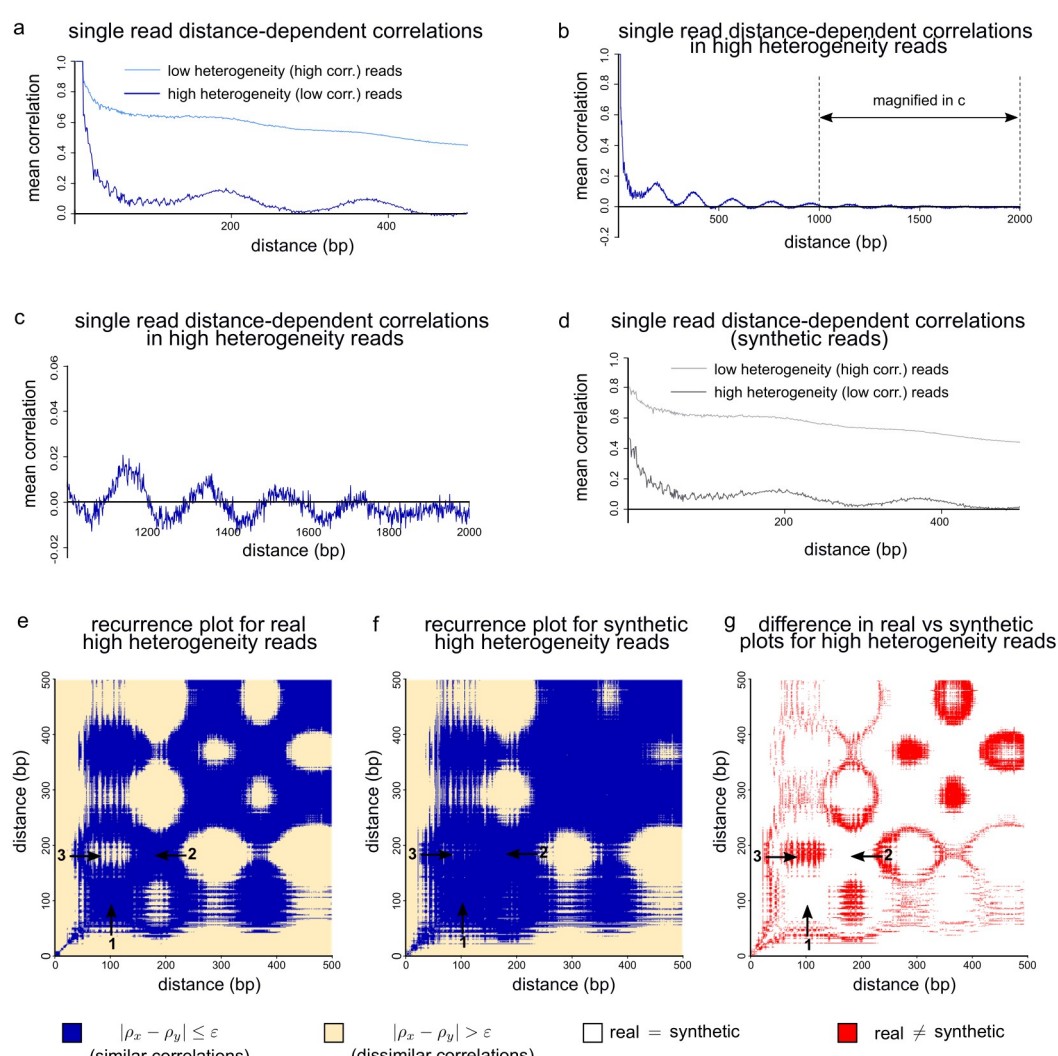

**Fig 5. Periodic DNA methylation patterns are observed in single molecules with high heterogeneity. a** Plot of mean single-read distance-dependent correlation between CpG sites within 500bp of each other for 1, 461, 561 high heterogeneity GM24385 reads and 544, 193 low heterogeneity GM24385 reads. **b** Plot of mean single-read distance-dependent correlation between CpG sites within 2kb of each other for 1, 461, 561 high heterogeneity GM24385 reads. **c** Magnification of the region indicated in **b** (i.e. the mean single-read distance-dependent correlation for distances 1kb-2kb in high heterogeneity GM24385 reads). **d** As in **a** but for synthetic reads generated based on bulk methylation properties. **e,f** Recurrence plots for real (**e**) and synthetic (**f**) high heterogeneity GM24385 reads for CpGs within 500bp of each other (see Methods). For these plots, $\varepsilon$ = 0.06. Arrows 1 and 2 indicate example regions where the correlations for distances $x$ and $y$ are similar in the real reads. Arrow 3 indicates an example region where the correlations for distances $x$ and $y$ are dissimilar in the real reads. **g** Difference plot derived from the recurrence plots in (**e**) and (**f**) showing agreement between the real and synthetic data in white and disagreement in red. The regions indicated by arrows 1 and 2 show agreement between the real and synthetic data. However the real and synthetic data disagree in the region indicated by arrow 3.

PMD reads showed oscillatory behaviour with wavelength in the range 176.8–198.0bp up to 2kb similarly to the high heterogeneity reads (S5d and S5e Fig).

We now aimed to determine the degree to which the single-molecule methylation patterns we observed were a consequence of the bulk mean methylation levels. Since the methylation state of each CpG in the single-molecule data is described using a binary variable, while the bulk mean methylation level of each CpG is a continuous variable, it is not possible to directly compare single-molecule states with bulk mean methylation levels. We therefore generated

synthetic single-molecule data based on the bulk GM24385 methylation data. In particular, for each read, we generated a synthetic read where the methylation state of each CpG in the synthetic read is determined by the probability that it is methylated in the bulk GM24385 data (see Methods). We then calculated distance-dependent correlations up to 500bp from these synthetic reads and compared them to what we observed in the real reads. The mean distance-dependent correlations derived from synthetic high heterogeneity reads showed oscillatory behaviour with wavelength in the range 177.5–198.3bp but the magnitude of these oscillations was significantly lower than that observed from the real high heterogeneity reads (Fig 5d, Wilcoxon ranked sign test, $p = 0.003$). This suggests that bulk methylation levels only partially explain the patterns observed in single molecules. As in the real low heterogeneity reads, no clear oscillatory behaviour was observed in the synthetic low heterogeneity reads (Fig 5d).

To further understand differences between the patterns observed within single-molecule high heterogeneity reads and those expected from mean bulk methylation levels, we used recurrence plots to examine distance-dependent correlations calculated up to 500bp for real and synthetic single reads. Recurrence plots use a binary colouring system which makes patterns within spatial and temporal data easily observable [30–32]. A recurrence plot is composed of a grid and, in the context presented here, block $(x, y)$ in the grid is one of two colours depending on whether the correlations associated with distances $x$ and $y$ are within a pre-specified tolerance of each other. The recurrence plot associated with the real high heterogeneity reads showed alternating areas of high and low similarity (Fig 5e). In particular, we observed large, regular blue areas corresponding to regions of high similarity in the correlations associated with different distances $x$ and $y$ (Fig 5e arrows 1 and 2). These had distinct boundaries from yellow elliptical regions corresponding to regions of low similarity (Fig 5e arrow 3). The recurrence plot associated with the synthetic high heterogeneity reads was similar. However, the observed structure in methylation patterns was less distinct than that of the real high heterogeneity reads, with correlations being generally more similar across distances (Fig 5f). Differences between the recurrence plots for the real and synthetic reads were concentrated at the boundaries separating regions where the correlations are similar from regions where the correlations are dissimilar (Fig 5g red regions). For example, the blue areas indicated by arrows 1 and 2 in Fig 5e–5g are larger in the recurrence plot associated with the synthetic reads in comparison to the recurrence plot associated with the real reads. This results in occlusion of the yellow elliptical region indicated by arrow 3 in the recurrence plot associated with the synthetic reads.

These analyses demonstrated that the parts of the human genome showing high intramolecular DNA methylation heterogeneity display kilobase-scale oscillatory DNA methylation patterns within single molecules that differ from those predicted from mean bulk DNA methylation levels.

## Periodic DNA methylation patterns at heterochromatin reflect nucleosome occupancy

The wavelength of the oscillations we observe in the high heterogeneity reads lies within the range of reported 160–230bp nucleosomal DNA repeat lengths [33] and it has been postulated that nucleosomes underpin oscillatory patterns observed in PMD bulk mean DNA methylation levels [27–29]. We therefore hypothesised that the oscillatory patterns that we observe in single molecules may reflect nucleosome positioning. To understand whether this might be the case, we examined nanopore sequencing of nucleosome occupancy and methylome (nanoNOMe-seq) data for the GM12878 cell line [34], which consisted of 21, 510, 519 autosomal reads with mean read length 10.5kb and mean autosomal reference CpG coverage 58.2. This

data combines native CpG methylation with exogenous GpC methylation introduced by the CviPI GpC methyltransferase, which preferentially methylates accessible DNA. Previous work has shown that nucleosomes are a major determinant of GpC methylation patterns in these assays due to the accessibility of GpCs in linker DNA [29]. To examine the relationship between DNA methylation and nucleosome positioning within single molecules, we called CpG and GpC methylation states across the genome from the nanoNOMe data and analysed their correlation within single reads. We omitted CpGs and GpCs within GCG contexts from the nanoNOMe analysis due to ambiguity as to whether methylation in these contexts arises due to native CpG methylation or exogenous GpC methylation [34].

We began by splitting nanoNOMe reads into a low and high heterogeneity group based on single-read CpG correlations as described above. As in the previous data sets analysed, we observed that the distance-dependent correlation between CpG sites was higher in the low heterogeneity reads in comparison to the high heterogeneity reads for all distances 2–500bp considered (Fig 6a). CpG oscillations with wavelength in the range 158.1–208.0bp were observed in high heterogeneity nanoNOMe reads but were less apparent in low heterogeneity nanoNOMe reads (Fig 6a), again agreeing with our previous analyses. To examine nucleosome occupancy patterns within single molecules we then calculated single-molecule distance-dependent correlations between GpC methylation states. We observed GpC oscillations with wavelength in the range 162.0–220.5bp in the high heterogeneity reads (Fig 6b). The similar wavelength of oscillations observed in the CpG and GpC data suggests that the periodic CpG methylation patterns observed in high heterogeneity reads could correlate with nucleosome positioning. We also observed similar GpC oscillations with wavelengths in the range 159.6–215.7bp in the low heterogeneity reads (Fig 6b), suggesting broadly similar nucleosome organisation in low and high heterogeneity genomic regions.

To more directly investigate the association between DNA accessibility and DNA methylation, we then calculated single-molecule distance-dependent correlations between CpG and GpC sites up to 500bp apart. These correlations showed oscillatory patterns with wavelength in the range 162.2–223.3bp in high heterogeneity reads (Fig 5c). We observed the CpG-GpC correlation to oscillate between positive and negative values in the high heterogeneity reads, with the positive peaks occurring at distances of approximately 1bp, 190bp and 380bp (Fig 5c). This suggests that CpG and GpC methylation tend to coincide on proximal bases. Given that nanoNOMe-seq results in GpC methylation being predominantly placed in linker DNA [29], this suggests that in high heterogeneity reads CpG methylation is predominantly found in linker DNA and excluded from nucleosomal DNA.

We also observed oscillatory patterns with wavelength in the range 153.9–216.0bp in the CpG-GpC correlation in low heterogeneity reads and the oscillations again suggest that CpG and GpC methylation states are most similar every $\sim$ 190bp. However, these correlations

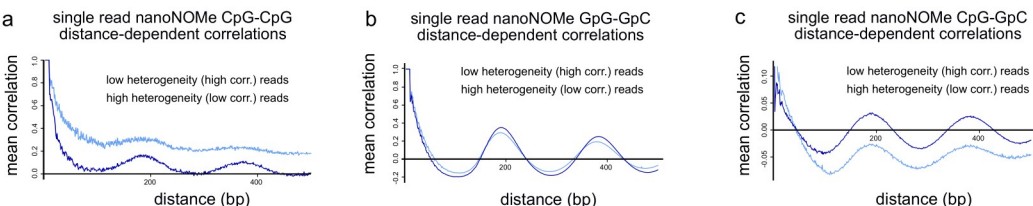

**Fig 6. Periodic DNA methylation patterns reflect chromatin accessibility. a** Plot of mean single-read distance-dependent correlation between CpG sites within 500bp of each other for 1, 373, 660 high heterogeneity nanoNOMe reads and 589, 154 low heterogeneity nanoNOMe reads. **b** As in **a** but for GpC correlations. **c** As in **a** but for correlations between CpG sites and GpC sites.

remain negative for distances greater than 45bp, indicating less association between DNA methylation and nucleosome occupancy in these reads (Fig 6c). The positive CpG-CpG correlation associated with low heterogeneity reads suggests high consistency in methylation state across these reads (Fig 6a). The periodicity observed in the CpG-GpC correlations in low heterogeneity reads is therefore likely to arise from the relationship between these largely uniform CpG methylation patterns and periodic GpC accessibility patterns (Fig 6b).

Overall these analyses suggest that nucleosome organisation is similar in low and high heterogeneity reads, but that the relationship between nucleosome positioning and DNA methylation differs between these groups.

We now sought to gain intuition as to whether nucleosome organisation at the single-molecule level could generate the differences in methylation patterns we previously observed between real reads and synthetic reads generated from bulk data (Fig 5g). At the single-molecule level, the length of linker DNA between nucleosomes can vary and nucleosomes can be differentially phased (positioned) between molecules [35]. We considered simplistic proof-of-principle models of extreme cases of nucleosome placement and interaction with DNA methylation on single molecules, where we assumed each nucleosome occupied 147bp of DNA, linker length varied between 10bp and 90bp, and methylation state was determined in a probabilistic manner based on nucleosome occupancy (see Methods).

We first generated single-molecule methylation data using a model where nucleosomes were identically phased between molecules (S6a Fig, see Methods) before calculating single-molecule distance-dependent correlations from this model and for modelled synthetic reads derived from its bulk properties. The single-molecule recurrence plot derived from the distance-dependent correlations displayed large, regular blue areas corresponding to regions of high similarity in the correlations associated with different distances $x$ and $y$ (S6b Fig arrows 1 and 2) which had distinct boundaries from yellow elliptical regions corresponding to regions of low similarity (S6b Fig arrow 3). This qualitatively captured the patterns we observed in the real high heterogeneity reads (S6b Fig vs. Fig 5e) and provides further evidence that nucleosomes could play a role in generating the oscillatory DNA methylation patterns we see in single molecules. In this case with perfect phasing between molecules, we observed that the recurrence plot generated using the bulk properties of the model was very similar to that of the modelled single-molecule data (S6b and S6c Fig).

We next generated data from a model where nucleosome positioning was completely unphased between molecules (S6d Fig, see Methods). We again observed large, regular blue areas corresponding to regions of high similarity in the correlations associated with different distances $x$ and $y$ (S6e Fig arrows 1 and 2) which had distinct boundaries from yellow elliptical regions corresponding to regions of low similarity (S6e Fig arrow 3). However, the recurrence plot generated using the bulk properties of the model showed correlations to be more similar across distances in comparison to the single-molecule recurrence plot generated from the model (S6e and S6f Fig). The blue areas indicated by arrows 1 and 2 in S6e and S6f Fig are larger in the recurrence plot associated with the synthetic reads in comparison to the recurrence plot associated with the real reads. This results in near complete occlusion of the yellow elliptical region indicated by arrow 3 in the recurrence plot associated with bulk properties of the model. Our observations from this model qualitatively agree with the differences we observed in the recurrence plots generated from real and bulk-simulated high-heterogeneity reads (Fig 5e–5g arrow 3), consistent with the hypothesis that these differences could be generated by differences in nucleosomal organisation between molecules.

Taken together, our analyses indicates that the periodic DNA methylation patterns we observe in high-heterogeneity reads reflects periodically positioned nucleosomes and that

differences between bulk and single-molecule methylation patterns could potentially arise from differences in nucleosome organisation between molecules.

## Discussion

Here, we have used Nanopore sequencing data to analyse large-scale intra-molecular DNA methylation heterogeneity in the human genome. Our results show that intra-molecular heterogeneity in DNA methylation patterns is abundant and underestimated by bulk-level metrics. We specifically observe a high degree of intra-molecular DNA methylation heterogeneity in heterochromatic regions that manifests as oscillatory DNA methylation patterns (Fig 7a). These patterns differ from those predicted from mean bulk methylation levels and reflect nucleosome occupancy.

We propose that the exclusion of DNA methylation from nucleosomal DNA is the most likely source of the heterogeneity we observe in reads arising from heterochromatin. The total length of nucleosomal and inter-nucleosomal linker DNA varies in the range of 160–230bp depending on the organism, cell type and region of the genome examined [33, 35, 40, 41]. *In vitro* experiments demonstrate that the wrapping of DNA around nucleosomes is refractory to methylation by DNMTs [36–39] so that methylation occurs at the linker DNA [42]. *In vivo* linker DNA methylation is seen in between the well-positioned nucleosomes around CTCF sites in human cells [43], consistent with the hypothesis that nucleosomes are also refractory to DNMTs *in vivo*. A similar organisation has been reported at CTCF sites at the single-molecule level [34]. This suggests that the blocking of DNMTs by nucleosomes could generate the oscillations that we observe in reads with high intra-molecular heterogeneity (Fig 7b).

However, only an estimated 8.7% of nucleosomes in the human genome are strongly-positioned [44] and the majority are not phased between molecules [35]. A single-molecule analysis of Micrococcal nuclease digested fragments suggests that heterochromatin is characterised by regularly spaced nucleosomal arrays [45]. Regular nucleosome spacing in heterochromatin was also observed in a single-cell nucleosomal positioning study which additionally suggested that heterochromatin is characterised by a lack of phasing between molecules [46]. In addition, a single-molecule footprinting analysis of oligonucleosomes released from cells found that irregular oligonucleosome patterns were abundant in heterochromatin [47]. Despite the reported lack of phasing of heterochromatic nucleosomes, analyses of bulk mean DNA methylation levels has reported structured periodicity in DNA methylation patterns in PMDs [27–

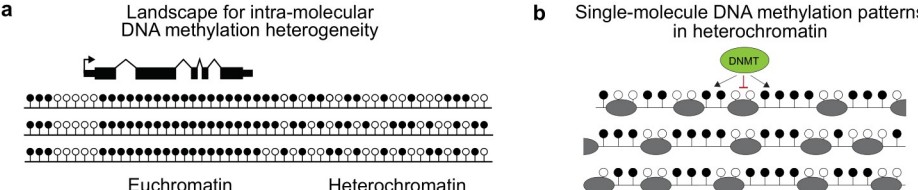

**Fig 7. Intra-molecular DNA methylation heterogeneity varies across the human genome and reflects nucleosome organisation at the single-molecule scale. a** Heterogeneous DNA methylation patterns are observed in heterochromatin. Schematic model of the arrangement of intra-molecular DNA methylation heterogeneity in the genome. Euchromatic, genic regions possess ordered, uniform patterns whereas heterochromatin is characterised by more heterogeneous patterns at the single-molecule level. **b** Oscillatory DNA methylation patterns are observed in single molecules within heterochromatin and could be caused by differences in nucleosomal organisation between molecules. Schematic showing proposed arrangement of DNA methylation and nucleosomes in single DNA molecules in heterochromatin. We propose that periodic patterns result from methylation of linker DNA due to nucleosomes being refractory to DNMTs [36–39]. Differences in nucleosome phasing and positioning between molecules could cause sites of methylation to be imperfectly aligned across molecules meaning that these patterns are partially masked in bulk DNA methylation data.

29]. Here we show for the first time that periodicity in heterochromatic DNA methylation patterns manifests at the kilobase scale within single molecules. We also find that the patterns we observe at the single-molecule level are not completely predicted from mean bulk DNA methylation levels; in fact they are partially masked. Our simple proof-of-principle models suggest that differences in the phasing and organisation of nucleosomes between molecules could potentially result in such masking (Fig 7b). We note that the oscillations previously observed in the distance-dependent correlations from heterochromatic bulk data suggest that heterochromatic nucleosomes show some degree of phasing and preferential positioning rather than being entirely unphased between molecules.

Our results also imply that there is a broadly similar nucleosome organisation in both euchromatic and heterochromatic regions of the genome. However, we only observe strongly periodic DNA methylation patterns in heterochromatic regions. This suggests a different relationship between nucleosomes and DNA methylation in euchromatin and heterochromatin. We suggest that the lack of oscillations observed in the low heterogeneity reads associated with euchromatic, genic regions could occur due to nucleosome remodelling. For example, the action of FACT in displacing nucleosomes during transcription [48] could enable DNMTs to access nucleosomal DNA.

We note that our analyses examine the correlation between accessibility and DNA methylation and cannot determine causal relationship between the DNA methylation and nucleosomes. It is possible instead that the patterns arise due to an influence of DNA methylation on nucleosome positions. For example, DNA methylation has been reported to affect the interaction of nucleosomes with a positioning sequence in the chicken beta-globin locus [49]. In addition, although nucleosome positioning is thought to be the major global influence on accessibility measured by CviPI methylation [29], other proteins can also affect accessibility, including transcription factors [50]. Further work will be required to precisely dissect the nature of the relationship between nucleosome occupancy and DNA methylation.

Previous work has also suggested that DNA sequence plays a strong role in shaping bulk DNA methylation patterns. The sequence immediately flanking a CpG is associated with, and can predict, PMD bulk mean methylation levels [27]. CpGs lacking neighbours and flanked by A or T bases have particularly low bulk mean methylation levels [51]. Such CpGs also lose methylation with successive cell divisions *in vitro* [52] and have been reported to have slower remethylation kinetics [53]. Deep-learning models can also accurately predict methylation loss based on the sequence of the 150bp surrounding a CpG [54]. However, nucleosome positioning is also non-deterministically influenced by sequence [55] and DNMT activity varies depending on CpG flanking sequences [56]. In addition, both modelling and experimental studies suggest that collaborative interactions between proximal CpG sites are required to maintain DNA methylation patterns [57–59], suggesting that local CpG density could also modulate DNA methylation patterns by reinforcing homogeneity in CpG dense regions. Collaborative methylation interactions are likely determined by the properties of the DNA methylation machinery. DNMT1 and DNMT3B can both methylate processively along DNA strands and DNMT3A forms multimers along the DNA fibre [60]. It is therefore possible that any associations existing between nucleosome positioning and DNA methylation could be linked with DNA sequence and CpG density, and future studies will be required to precisely dissect their respective roles.

The accuracy of results obtained from the analysis of methylation data is dependent upon the technology used to detect methylation. While we used Nanopolish to call methylation from Nanopore data other methylation callers exist including Guppy, Megalodon, Remora and Dorado from Nanopore, as well as other callers including DeepSignal, Tombo, METEORE and DeepMod [61–64]. Benchmarking studies such as [65, 66] have found Nanopolish to be amongst the top-performing methylation callers, both in terms of accuracy and computational

efficiency, though such studies have not assayed performance at the single-molecule level. To deal with erroneous methylation calls within single molecules, we filter out low-confidence Nanopolish methylation calls. However, we note that such filtering may bias against certain motifs and sequence contexts. We also restrict our attention to single-molecule patterns consisting of a minimum of 100 called CpGs, meaning the impact of a small number of miscalled CpGs when calculating read statistics should be minimal. In addition, Nanopolish is unable to distinguish methylation states between CpGs that lie in very close proximity of each other, leading to the same methylation state being called for CpGs within 10bp of each other. It is possible that this is why we do not observe the 10bp periodicity in the distance-dependent correlation previously observed from bulk bisulfite sequencing data that has been postulated to be representative of DNA winding around the DNA helix [27]. The inability of Nanopolish to distinguish methylation states of CpGs that are very close together could potentially lead to over-estimation of the correlation between neighbouring CpGs in high-density regions of the genome. Nonetheless, an over-estimation in the correlation between neighbouring CpGs would result in an under-estimation of heterogeneity. This means that the high heterogeneity we observe in heterochromatin and PMDs is unlikely be the product of such errors. Improved methylation calling will likely allow additional insight into the nature of single-molecule DNA methylation patterns.

A variety of statistics have previously been used to examine heterogeneity in DNA methylation patterns. These include the proportion of discordant reads (PDR) [67], methylation haplotype load (MHL) [68] and fraction of discordant reads (FDR) [20]. However, these measures quantify inter-molecular heterogeneity and so are not directly comparable to our analysis of intra-molecular heterogeneity. An information theory approach has been used to predict probability distributions associated with single-molecule methylation patterns from WGBS data and suggested that the genome is organised into domains of disordered and ordered methylation patterns that coincide with large-scale chromatin organisation [10]. However, although these studies are consistent with our observation of high intra-molecular heterogeneity in heterochromatin, WGBS reads are not sufficiently long to directly assess single-molecule DNA methylation patterns across the scale of $10^4$-$10^5$ bases as we have done here.

## Conclusion

Taken together our observations suggest that the degree of intra-molecular heterogeneity in DNA methylation patterns varies widely across the human genome. We propose that the nature of this heterogeneity suggests a role for nucleosomes in shaping DNA methylation patterns at the single-molecule level, particularly in heterochromatin.

## Materials and methods

Unless otherwise specified, we performed analysis using R (v. 4.1.2), Mathematica (v. 12.3) and Bedtools (v. 2.30.0) [69]. All statistical tests were two-sided.

### Data sources

We downloaded raw Nanopore reads in fast5 format for GM24385 lymphoblastoid cells from Oxford Nanopore (amazon bucket s3://ont-open-data/gm24385_2020_09/). Fast5 Nanopore reads were also downloaded for the GM12878 cell line [19] (amazon bucket s3://nanopore-human-wgs/rel6/MultiFast5Tars/) and used to replicate key analyses. Finally, we downloaded GM12878 nanoNOMe fast5 files [34] (https://trace.ncbi.nlm.nih.gov/Traces/index.html?view=study&acc=SRP173931). We refer to the GM12878 nanoNOMe data as simply the nanoNOMe data to avoid confusion with the other GM12878 Nanopore dataset analysed.

## Preprocessing of Nanopore data

We basecalled the GM24385 and GM12878 reads using the dna_r9.4.1_450bps_hac_prom.cfg and dna_r9.4.1_450bps_hac.cfg configurations of Guppy (v. 5.0.11) respectively [70]. We basecalled the nanoNOMe data generated using minION and gridION flow cells using the dna_r9.4.1_450bps_hac.cfg configuration of Guppy (v. 5.0.11) and basecalled the nanoNOMe data generated using promethION flow cells using the dna_r9.4.1_450bps_hac_prom.cfg configuration of Guppy (v. 5.0.11). We then aligned these reads to the human genome (hg38 assembly) using the map-ont setting of Minimap2 (v. 2.22) [71] and called the methylation state at CpGs using Nanopolish call-methylation with default settings (v. 0.13.2) [18]. Methylation states were assigned to each CpG site based on the log likelihood ratio (LLR) calculated by Nanopolish. We used the LLR cut-off used by Nanopolish scripts and estimated to have an accuracy >92.5% [18]. Comparisons to WGBS also suggest Nanopolish methylation calls are accurate [65, 66]. Here we consider single-molecule patterns $\geq$100 CpGs in length so the influence of a small number of miscalled methylation states is likely to be minimal. CpG sites were assigned an unmethylated status if LLR $\leq -2$ and a methylated status if LLR $\geq 2$. For |LLR|$< 2$ we deemed the CpG site to be unassignable. Using these settings, 76.0% of autosomal CpG observations were assigned methylated or unmethylated states in the GM24385 data (69.2% in the GM12878 data and 68.9% in the nanoNOMe data). For the nanoNOMe data we additionally called the methylation state of GpC sites using Nanopolish call-methylation with –q gpc and default settings (v. 0.13.2), where each GpC site was again deemed to be methylated if LLR$\geq 2$, unmethylated if LLR$\leq -2$ and unassigned if |LLR| $< 2$ leading to 91.6% of autosomal GpC sites being assigned a methylated or unmethylated status. For the nanoNOMe data we excluded both CpG and GpC sites within GCG contexts due to the uncertainty in whether methylation in these contexts arises from native CpG methylation or exogenous GpC methylation. Only reads mapping to canonical hg38 autosomes (chromosomes 1–22) were considered in our analyses.

## Calculation of single-read statistics

We converted methylation tsv files outputted by Nanopolish to bed format using the mtsv2bedGraph.py script from the timplab/nanopore-methylation-utilities GitHub repository [34]. We used a custom R script to identify the 2, 908, 181 out of 6, 289, 480 reads in the GM24385 data that had defined methylation states for $\geq$100 CpGs (1, 331, 213 out of 10, 784, 132 reads in the GM12878 data and 2, 648, 588 out of 21, 510, 519 reads in the nanoNOMe data). For each retained read, $r$, we discarded CpG sites that could not be assigned a methylation state and stored the methylation states of the remaining $N_r \geq 100$ CpGs in a vector $v_r = \{v_{r,1}, v_{r,2}, \ldots, v_{r,N_r}\}$, where for $i = 1, \ldots . N_r$,

$$v_{r,i} = \begin{cases} 1, & \text{if CpG } i \text{ in read } r \text{ is methylated,} \\ 0, & \text{if CpG } i \text{ in read } r \text{ is unmethylated.} \end{cases} \tag{1}$$

For each read $r$ we then used R (v. 4.0.1) to calculate the mean methylation level ($\mu_r$), coefficient of variation (CV$_r$) and the correlation in methylation states between neighbouring sites ($\rho_r$) via

$$\mu_r = \frac{1}{N_r} \sum_{i=1}^{N_r} v_{r,i}, \tag{2}$$

$$\text{CV}_r = \frac{\sigma_r}{\mu_r}, \quad \text{where} \quad \sigma_r^2 = \frac{1}{N_r - 1} \sum_{i=1}^{N_r} (v_{r,i} - \mu_r)^2, \tag{3}$$

$$\rho_r = \frac{\sum_{i=1}^{N_r - 1} (v_{r,i} - \hat{\mu})(v_{r,i+1} - \tilde{\mu})}{\sqrt{\sum_{i=1}^{N_r - 1} (v_{r,i} - \hat{\mu})^2 \sum_{i=2}^{N_r} (v_{r,i} - \tilde{\mu})^2}}, \tag{4}$$

where

$$\hat{\mu} = \frac{1}{N_r - 1} \sum_{i=1}^{N_r - 1} v_{r,i} \quad \text{and} \quad \tilde{\mu} = \frac{1}{N_r - 1} \sum_{i=2}^{N_r} v_{r,i}.$$

### Definition of genomic windows

We split canonical autosomal chromosomes into 28,760 non-overlapping genomic windows of 100kb using BEDOPS (v. 2.4.26) [72]. Note that 22 windows (corresponding to the end of each autosome) were shorter than 100kb. This process was repeated but with 50kb and 200kb windows for selected analyses.

### Calculation of mean single-read statistics

We used Bedtools and custom command line scripts to extract the 1, 707, 118 GM24385 reads (1, 072, 713 GM12878 reads) that were entirely contained within a single window and had called methylation states for $\geq 100$ CpGs. We then calculated mean single-read statistics associated with each window using R (v. 4.0.1). In particular, if $r_1, r_2, \ldots, r_{N_w}$ were the reads that aligned entirely within a window, $n$, then we calculated the single-molecule mean ($\mu_{s,n}$), coefficient of variation ($\text{CV}_{s,n}$) and correlation ($\rho_{s,n}$) associated with window $n$ via

$$\mu_{s,n} = \frac{1}{N_w} \sum_{r=1}^{N_w} \mu_r, \qquad \text{CV}_{s,n} = \frac{1}{N_w} \sum_{r=1}^{N_w} \text{CV}_r, \qquad \rho_{s,n} = \frac{1}{N_w} \sum_{r=1}^{N_w} \rho_r. \tag{5}$$

This lead to mean single-molecule statistics being obtained for 26, 951 GM24385 100kb windows (26, 782 GM12878 100kb windows).

### Calculation of bulk-level window statistics

For each dataset we defined the methylation coverage for each CpG site to be the combined number of methylated and unmethylated calls associated with that CpG site. We discarded CpG sites with <10 methylation calls and those that overlapped the boundary between two windows (i.e. if the forward-strand cytosine belongs to one window and the guanine to another). The mean CpG methylation for each site was defined as the number of methylated calls associated with that CpG divided by the total number of calls (methylated and unmethylated) observed for that CpG. We then removed 100kb windows with <100 retained CpG sites. We used the CpG methylation levels associated with the remaining 26, 714 windows in the GM24385 data (26, 565 windows in the GM12878 data) to calculate bulk-level statistics for each window in R (v. 4.0.1). In particular, for window $n$ with $N \geq 100$ retained CpGs with

methylation levels $m_1, m_2, \ldots, m_N$, we calculated the bulk mean methylation level ($\mu_{b,n}$), coefficient of variation ($CV_{b,n}$) and the correlation in methylation level between nearest-neighbour sites ($\rho_{b,n}$) via

$$\mu_{b,n} = \frac{1}{N} \sum_{i=1}^{N} m_i, \tag{6}$$

$$CV_{b,n} = \frac{\sigma_{b,n}}{\mu_{b,n}}, \quad \text{where} \quad \sigma_{b,n}^2 = \frac{1}{N-1} \sum_{i=1}^{N} (m_i - \mu_{b,n})^2, \tag{7}$$

$$\rho_{b,n} = \frac{\sum_{i=1}^{N-1} (m_i - \hat{\mu}_n)(m_{i+1} - \tilde{\mu}_n)}{\sqrt{\sum_{i=1}^{N-1} (m_i - \hat{\mu}_n)^2 \sum_{i=2}^{N} (m_i - \tilde{\mu}_n)^2}}, \tag{8}$$

where

$$\hat{\mu}_n = \frac{1}{N-1} \sum_{i=1}^{N-1} m_i \quad \text{and} \quad \tilde{\mu}_n = \frac{1}{N-1} \sum_{i=2}^{N} m_i.$$

## Comparison of mean single-molecule and bulk-level window statistics

We used Spearman correlation tests in R to calculate correlations and associated p-values between the bulk statistics and mean single-molecule statistics for the 26,687 100kb genomic windows in the GM24385 data that had sufficient bulk and single-molecule data. We also fit linear models to the bulk-level and single-molecule statistics using the base lm command in R and conducted paired t-tests to assess the difference between the bulk and single-molecule values. The average absolute relative difference between the bulk and single-molecule mean was calculated via

$$\frac{1}{26,687} \sum_{n=1}^{26,687} \frac{|\mu_{b,n} - \mu_{s,n}|}{\mu_{s,n}}. \tag{9}$$

Similarly, the average absolute relative difference between the bulk and single-molecule coefficient of variation and correlation were calculated via

$$\frac{1}{26,687} \sum_{n=1}^{26,687} \frac{|CV_{b,n} - CV_{s,n}|}{CV_{s,n}} \quad \text{and} \quad \frac{1}{26,687} \sum_{n=1}^{26,687} \frac{|\rho_{b,n} - \rho_{s,n}|}{\rho_{s,n}}, \tag{10}$$

respectively.

## Generation of simulated and shuffled correlations from individual reads

We simulated the case where methylation states were randomly distributed along each read. For each read $r$ in the GM24385 data with methylation states for $\geq 100$ CpGs, we created a simulated read containing the same number of CpGs, but where the methylation state of each CpG was randomly chosen from a binomial distribution with success rate for methylation equal to the mean methylation level of the read ($\mu_r$). The correlation was then calculated for

each of these synthetic reads. The correlation distribution obtained from the true reads was compared to that obtained from the synthetic reads and a Kolmogorov-Smirnov test was conducted in R to assess whether the observed methylation patterns in true reads are random or non-random.

To generate the mean shuffled correlation for genomic windows we calculated a mean correlation for each window using random GM24385 reads (rather than those that align to that window). Reads were shuffled using the command-line *shuf* command. For example, if $\{r_1, r_2, \ldots, r_{N_w}\}$ were the reads aligning entirely within window *n*, then we replaced these by a set of random reads $\{\hat{r}_1, \hat{r}_2, \ldots, \hat{r}_{N_w}\}$, and calculated a mean correlation for window *n* via

$$\widehat{\rho}_{s,n} = \frac{1}{N_w} \sum_{i=1}^{N_w} \rho_{\hat{r}_i}.$$

We then compared the $\rho_{s,n}$ and $\hat{\rho}_{s,n}$ distribution using a Kolmogorov-Smirnov test in R.

## Calculation of RTS

For each read *r* containing methylation states for $\geq 100$ CpGs, we calculated the read transition score in R (v. 4.0.1) as

$$\text{RTS}_r = \frac{1}{N-1} \sum_{i=1}^{N-1} |v_{r,i+1} - v_{r,i}|. \tag{11}$$

$\text{RTS}_r$ essentially measures the probability that two nearest-neighbour CpG sites in read *r* have different methylation states. Reads which are entirely unmethylated or entirely methylated therefore have RTS 0, while reads where no two nearest-neighbour CpG sites share the same methylation state have RTS 1.

Mean RTSs were then calculated for each genomic window similarly as for the other mean single-read statistics. In particular, if $\{r_1, r_2, \ldots, r_{N_w}\}$ were the reads that aligned entirely within a window, *n*, then we took the mean RTS associated with window *n* to be

$$\text{RTS}_n = \frac{1}{N_w} \sum_{i=1}^{N_w} \text{RTS}_{r_i}. \tag{12}$$

Shuffled mean RTSs were also obtained for each window in a similar manner as shuffled read correlations.

## Defining windows with the lowest and highest intra-molecular heterogeneity

We ranked the 26, 951 GM24385 windows (26, 782 GM12878 windows) by $\rho_{s,n}$ before defining the windows with the greatest intra-molecular heterogeneity as the 10% of windows with the lowest mean correlation and the least heterogeneous windows as the 10% of windows with the highest mean correlation.

## Comparison of intra-molecular heterogeneity levels to inter-molecular heterogeneity levels

We computed the measure $d_i = |m_i - 0.5|$, where $m_i$ is the methylation level of CpG *i* for each CpG with $\geq 10$ methylation calls. $d_i = 0.5$ indicates low molecule-to-molecule heterogeneity for CpG *i*, where as $d_i = 0$ corresponds to high molecule-to-molecule heterogeneity for CpG *i*. We

performed Kolmogorov-Smirnov tests to compare the distribution of $d_i$ associated with the 10% of windows with the lowest mean correlation and the distribution associated with the rest of the genome and to compare the 10% of windows with the highest mean correlation and the distribution associated with the rest of the genome. In particular, we used Kolmogorov-Smirnov tests in R to test whether the cumulative distribution function (CDF) associated with $d_i$ for the lowest or highest mean correlation windows is significantly different than that associated with the rest of the genome.

To test the relationship between mean correlation and mean $d_i$ value, we calculated the Spearman correlation between the mean correlation and the mean $d_i$ in each window using R.

## Sourcing and processing of genomic annotations

Using the command line, we derived bed files of hg38 gene locations using annotation files from Gencode (v. 41) [73]. CGI annotations were taken from [74]. Overlapping CGI intervals were merged using BEDtools (v. 2.27.1) before they were converted to hg38 positions using the UCSC browser liftover tool (https://genome.ucsc.edu/cgi-bin/hgLiftOver). Non-autosomal CGIs were excluded from the analysis as were CGIs located in ENCODE blacklist regions [23] as previously described [75].

We downloaded hg19 GM12878 lymphoblastoid cell chromHMM states from UCSC (http://www.genome.ucsc.edu/cgi-bin/hgTables), [23, 24] before lifting them over to hg38 genome coordinates (https://genome.ucsc.edu/cgi-bin/hgLiftOver). To remove cases where multiple hg19 coordinates mapped to the same hg38 coordinate, we used the command line and Bedtools to remove from the analysis any regions associated with more than one annotation type in hg38 as a consequence of this liftover.

We downloaded PMD locations identified for 11 lymphoblastoid cell lines from the AllPMDs.tar.bz2 file in the bdecatoPMD_Paper_Scripts GitHub repository [26]. These coordinates were then converted from hg19 to the hg38 coordinates using Liftover (https://genome.ucsc.edu/cgi-bin/hgLiftOver). We excluded poorly mapped regions of the genome using annotations of gaps and centromeres from the UCSC browser (hg38 gap and centromere tracks). Annotations were downloaded from the UCSC table browser. Regions annotated as heterochromatin, short arm and telomeres from the gaps track were merged with the centromeres track using Bedtools merge with -d set to 10Mb. This merged file was then excluded from the PMD BED files using Bedtools subtract.

## Definition and analysis of PMDs

We called PMDs for GM24385 and GM12878 cells using Methpipe (v. 5.0.0) [26] on mean bulk CpG methylation levels and CpG methylation coverage calculated from Nanopore data. We defined non-PMDs as genomic regions that did not overlap with a PMD. Poorly mapped regions of the genome were removed from the PMD and non-PMD annotations as previously discussed. Using Bedtools we also excluded PMDs and non-PMDs shorter than 200kb. We compared the overlap between GM24385 PMDs to those from other lymphoblastoid cells [26] using the Bedtools Jaccard function on a BED file generated by merging the PMDs annotated in the other cell lines using Bedtools merge.

## Testing enrichment of annotations in genomic windows

For each annotation type, $a$, we used Bedtools to compute

- $P_{a,\text{all}}$: the proportion of bases in all windows analysed that overlap the annotation (i.e. the sum of bases from all windows that overlap the annotation divided by the summed length of all windows);

- $P_{a,\text{least}}$: the proportion of bases in the least heterogeneous windows that overlap the annotation (i.e. the sum of bases from the least heterogeneous windows that overlap the annotation divided by the summed length of the least heterogeneous windows);

- $P_{a,\text{most}}$: the proportion of bases in the most heterogeneous windows that overlap the annotation (i.e. the sum of bases from the most heterogeneous windows that overlap the annotation divided by the summed length of the most heterogeneous windows).

We then calculated the enrichment of the least and most heterogeneous windows in annotation type $a$ using

$$e_{a,\text{least}} = P_{a,\text{least}}/P_{a,\text{all}} \quad \text{and} \quad e_{a,\text{most}} = P_{a,\text{most}}/P_{a,\text{all}},$$

respectively. To test the significance of the enrichment, we extracted the proportion that each window overlaps annotation type $a$ using Bedtools and compared the distribution of proportions associated with the least/most heterogeneous windows to the proportions associated with all windows using Wilcoxon tests in R.

## Testing continuous relationship between correlation and annotations

To test the continuous relationship between mean correlation in a window and the proportion of overlap with genomic annotations, we calculated the correlation between mean correlation in a window and its overlap with heterochromatin and PMDs using R. For graphing, we split windows into five equally sized groups based on their ranked mean correlation. We then repeated the enrichment analyses previously described to calculate the annotation enrichment level in each group.

To test the relationship between correlations within single reads and overlap with genomic annotations, we used Bedtools to extract the proportion of each read that overlapped the chromHMM heterochromatin state or PMDs and conducted Spearman correlation tests in R to compare these overlaps to the read correlations.

## Derivation of low and high correlation reads by mixture modelling

Using Mathematica we fit the sum of two Gaussian distributions, with means $\mu_1, \mu_2$ ($\mu_1 < \mu_2$) and standard deviations $\sigma_1, \sigma_2$, to the bimodal distribution for the read correlation (Fig 2a). Of the 2, 907, 431 GM24385 reads (respectively, 1, 327, 908 GM12878 reads and 2, 648, 457 nano-NOMe reads) containing methylation information for a minimum of 100 CpG sites and for which the correlation between nearest-neighbour sites could be computed, we defined the 1, 461, 561 (respectively, 714, 789 and 1, 373, 660) reads that satisfied $|\rho_r - \mu_1| \leq \sigma_1$ to be the "low correlation group" (or "high heterogeneity group") and the 544, 193 (respectively, 205, 583 and 589, 154) reads that satisfied $|\rho_r - \mu_2| \leq \sigma_2$ to be the "high correlation group" (or "low-heterogeneity group"), where $\rho_r$ is the correlation associated with read $r$. We discarded the remaining reads. The previous enrichment analyses were then repeated for the low and high correlation read groups.

## Calculating distance-dependent correlations

For each read in the high heterogeneity (i.e. low correlation) group, we extracted the methylation state of each possible pair of CpGs along with the distances between those CpGs. In

particular, for a read $r$ covering $N_r$ CpG sites, we extracted $\{u_i, u_j, d_{i,j}\}$ for each $i \in \{1, 2, \ldots, N_r - 1\}$, $j \in \{i + 1, \ldots, N_r\}$, where

$$
u_i = \begin{cases}
1, & \text{if CpG } i \text{ in read } r \text{ is methylated,} \\
0, & \text{if CpG } i \text{ in read } r \text{ is unmethylated,} \\
\text{NA}, & \text{if CpG } i \text{ in read } r \text{ is unassigned,}
\end{cases}
$$

and $d_{i,j}$ is the distance (in bps) between CpG $i$ and CpG $j$ in read $r$. For each distance $d = \{2, 3, \ldots, 2000\}$ we extracted $\{u_i, u_j\}$ corresponding to $d_{i,j} = d$, $u_i \neq \text{NA}$ and $u_j \neq \text{NA}$ to obtain a matrix of states $S$ with $L \geq 0$ rows, where each row denotes the methylation state of a pair of CpGs lying a distance of $d$ apart from each other in read $r$. If $L \geq 2$, then we used R (v. 4.0.1) to compute the correlation between sites on read $r$ a distance of $d$ apart via

$$
\rho_{r,d} = \frac{\sum_{i=1}^{L}(S_{i,1} - \mu_1)(S_{i,2} - \mu_2)}{\sqrt{\sum_{i=1}^{L}(S_{i,1} - \mu_1)^2 \sum_{i=1}^{L}(S_{i,2} - \mu_2)^2}},
$$

where

$$
\mu_1 = \frac{1}{L}\sum_{i=1}^{L} S_{i,1} \quad \text{and} \quad \mu_2 = \frac{1}{L}\sum_{i=1}^{L} S_{i,2}.
$$

For each $d = \{2, 3, \ldots, 2000\}$ we computed the mean of $\rho_{r,d}$ over all reads in the high heterogeneity group to obtain mean single-molecule distance-dependent correlations associated with the high heterogeneity group.

We obtained mean single-molecule distance-dependent correlations associated with the low heterogeneity group analogously for distances 2–500bp.

Reads aligning entirely within PMDs were also extracted using Bedtools and mean single-molecule distance-dependent correlations associated with these reads were calculated for distances 2–2000bp as above.

For the low and high heterogeneity nanoNOMe reads, we also calculated GpC-GpC distance-dependent correlations and CpG-GpC distance-dependent correlations up to 500bp in an analogous manner to the CpG-CpG correlations.

We obtained the power spectrum associated with distance-dependent correlations up to 2kb for the high heterogeneity GM24385 reads using the PowerSpectralDensity function in Mathematica and observed two prominent peaks. One of these peaks was associated with a frequency close to zero and suggested the presence of oscillations of wavelength over 3kb. Since only distances up to 2kb are included in the correlation data this non-physical peak was discarded. We then used the Maximise command in Mathematica to extract the value of the remaining peak, $PS_{\max}$, the frequency at which this peak was attained, $\omega_{\text{peak}}$ and frequency range surrounding this peak for which the power spectrum takes at least half the value of $PS_{\max}$, $(\omega_{\min}, \omega_{\max})$. We then calculated the spectral range associated with this peak via $\{\frac{2\pi}{\omega_{\max}}, \frac{2\pi}{\omega_{\min}}\}$ and took this to be the range of wavelengths associated with the peak. We calculated a range for wavelengths associated with oscillations observed in other distance-dependent correlations in a similar manner.

## Generation of synthetic reads from bulk DNA methylation data

We edited the calculate_methylation_frequency.py script provided by Nanopolish [18] to output the proportion of times that each CpG site was called to be unmethylated, methylated or unassigned across reads, where again each CpG site was called to be methylated if LLR$\geq$ 2, unmethylated if LLR$\leq$ −2 and unassigned if |LLR| < 2. For each read in the GM24385 dataset we then generated a synthetic read based on this bulk information. Explicitly, for each read $r$, a synthetic read $\hat{r}$ was created where the methylation state of CpG $i$ in read $\hat{r}$ is set to be unmethylated with probability $\mu_{u,i}$, methylated with probability $\mu_{m,i}$ and unassigned with probability $\mu_{\mathrm{NA},i}$, where $\mu_{u,i}$, $\mu_{m,i}$, $\mu_{\mathrm{NA},i}$ are the bulk probabilities that CpG $i$ is unmethylated, methylated or unassigned.

The synthetic reads corresponding to the low and high heterogeneity reads were then extracted and used to calculated distance-dependent correlations as previously.

## Generation of recurrence plots

Recurrence plots were generated using the Mathematica ListDensityPlot command. In the recurrence plots, block $(x, y)$ was set to be blue if $|\rho_x - \rho_y| \leq \varepsilon$ for some $\varepsilon > 0$, where $\rho_x$ and $\rho_y$ are the correlations associated with distances $x$ and $y$, respectively, and is yellow otherwise. We chose $\varepsilon = 0.06$ as this was the tolerance value for which the Euclidean distance between the real and synthetic recurrence plots is maximised.

## Mathematical models of nucleosome occupancy

We generated simple models of the interaction between DNA methylation and nucleosomes using Mathematica. If methylation state was strongly dictated by nucleosome occupancy, then DNA methylation data would sample nucleosome occupancy data where CpG sites are present. We therefore simplified our models and made them more tractable with a lower number of simulated reads by assuming that every DNA base on the molecules could be methylated. As nucleosomes are refractory to the action of DNMTs, we assumed that each nucleosome occupied DNA base was unmethylated with probability $p = 0.8$ and each linker DNA base was methylated with probability $p = 0.8$ [36–39, 42]. We also assumed that nucleosomes occupied 147bp of DNA and that linker length varied uniformly within the range 10 to 90bp [33]. We simulated two mathematical models of nucleosome occupancy. In each model, DNA methylation data was simulated for $m = 200$ reads ($R_1, \ldots, R_{200}$) covering the same region of length $n = 2000$bp.

To generate the first read, $R_1$, in each model we first randomly selected the start position of the first nucleosome from the first 90bp of the read. The nucleosome occupancy status of each bp in the read was then simulated under the assumptions that each nucleosome occupies 147bp of DNA and that the length of linker DNA between nucleosomes can take any value between 10bp and 90bp with equal probability. Each nucleosome occupied DNA base was taken to be unmethylated with probability $p = 0.8$, while each unoccupied base was taken to be methylated with probability $p = 0.8$. For $i = 1, \ldots, 2000$, we set $R_{1,i} = 0$ if base $i$ was unmethylated and $R_{1,i} = 1$ if base $i$ was methylated.

The first model considered the case where nucleosomes are perfectly phased between molecules, i.e. nucleosomes are positioned in exactly the same place in each molecule. However, we note that while each read in this model was associated with the same nucleosome occupancy pattern as $R_1$, the associated methylation pattern varied from read to read due to the probabilistic way in which methylation state was assigned to occupied and unoccupied DNA bases. In the second model, it is assumed that there is no phasing of nucleosomes between molecules, i.e. nucleosome positioning in one molecule is entirely independent of nucleosome positioning

in other molecules. In this case, each read $R_2, \ldots, R_{200}$ was simulated independently, using the same approach as used to simulate $R_1$.

For each model, we used the simulated data to calculate single-molecule distance-dependent correlations in methylation status. In particular, for $d = 1, \ldots, 500$, the correlation in methylation state associated with bases a distance $d$bps apart was calculated for read $i$ via

$$\rho_{i,d} = \frac{\sum_{j=1}^{2000-d} (R_{i,j} - \hat{\mu})(R_{i,j+d} - \tilde{\mu})}{\sqrt{\sum_{j=1}^{2000-d} (R_{i,j} - \hat{\mu})^2 \sum_{i=1+d}^{2000} (R_{i,j} - \tilde{\mu})^2}}, \tag{13}$$

where

$$\hat{\mu} = \frac{1}{2000-d} \sum_{j=1}^{2000-d} R_{i,j} \quad \text{and} \quad \tilde{\mu} = \frac{1}{2000-d} \sum_{j=1+d}^{2000} R_{i,j}.$$

We then used the simulated data to derive bulk-level DNA methylation data. In particular, we calculated the bulk probability that base $j \in \{1, 2, \ldots, 2000\}$ is methylated or unmethylated, respectively, via

$$\mu_{meth,j} = \sum_{i=1}^{200} \frac{R_{i,j}}{200} \quad \text{and} \quad \mu_{unmeth,j} = 1 - \mu_{meth,j}.$$

We then used this information to create "simulated" synthetic reads $(\hat{R}_1, \ldots, \hat{R}_{200})$ for each model in an analogous way as was done for the real data. Finally, we used the simulated synthetic reads to calculate distance-dependent correlations in methylation state in a similar way as described above.

## Supporting information

**S1 Fig. Large-scale methylation heterogeneity is underestimated by bulk analysis of methylomes independently of the window size used.** Density scatter plots of mean single-read statistics vs. bulk statistics for 50kb genomic windows ($n = 52, 840$) (**a**) and 200kb genomic windows ($n = 13, 439$) (**b**). Left: mean methylation level, middle: coefficient of variation, right: correlation between neighbouring CpG sites. Dashed and solid lines show lines of identity and linear models fitted to the data respectively. Spearman correlations are shown, as are p-values from paired t-tests comparing bulk to single-read statistics.
(TIF)

**S2 Fig. Intra-molecular heterogeneity varies across the human genome. a** Histogram showing the correlation distribution for the 1, 331, 213 GM12878 reads containing methylation states for $\geq 100$ CpGs. **b** Histogram of the mean correlation distribution in GM12878 for 100kb genomic windows ($n = 26, 782$), calculated using reads that align entirely within each window. **c** Schematic of RTS values calculated for hypothetical 10 CpG reads. **d** Histogram showing the RTS distribution for the 2, 908, 181 individual GM24385 reads containing methylation information for $\geq 100$ CpGs. **e** Density scatter plot of the RTS vs. correlation for individual GM24385 reads. **f** Density scatter plot of the mean RTS vs. mean correlation for 100kb genomic windows in GM24385. In **e, f** the solid line shows a linear model fitted to the data and the Spearman correlation coefficient is shown in the top right. **g** Histogram showing the distribution of mean RTS in GM24385 for 100kb windows ($n = 26, 951$) calculated using reads

that align entirely within these windows. **h** Histogram showing the mean shuffled RTS in GM24385 for 100kb windows ($n = 26, 951$), calculated using random reads rather than those that align to each window.
(TIF)

**S3 Fig. Equivalent levels of intra-molecular heterogeneity can be associated with different levels of inter-molecular heterogeneity. a** Shown are six hypothetical examples of genomic windows composed of 10 CpG sites and covered by four reads. For each example, the correlation associated with each read is shown to the right of the read and the mean correlation for the window is shown below each example. The mean bulk methylation of each individual CpG is indicated below the reads. Each pair of examples: windows 1 and 2, windows 3 and 4, windows 5 and 6, show situations where the same mean window correlation is associated with differing degrees of inter-molecular heterogeneity. **b** Density scatter plot of mean window correlation vs. mean window $d_i$ for all 100kb genomic windows. Spearman correlation (Rho) is shown along with the corresponding p-value.
(TIF)

**S4 Fig. Heterogeneous genomic windows are enriched in heterochromatin and PMDs independently of the window size used. a** Barplots showing enrichments of the least and most heterogeneous GM24385 50kb windows in GM12878 chromHMM states. **b** Barplots showing enrichment of the least and most heterogeneous GM24385 200kb windows in GM12878 chromHMM states. **c** Barplots showing enrichment of the least and most heterogeneous GM12878 100kb windows in GM12878 chromHMM states. **d** Genome browser plot showing a representative genomic region with the mean DNA methylation and mean correlation in 100kb windows alongside GM24385 PMD locations and those previously identified in 11 other lymphoblastoid cell lines [26]. **e** Barplots showing enrichment of the least and most heterogeneous GM24385 50kb windows (left) and 200kb windows (right) in PMDs and non-PMDs. Shown are significant p-values from Wilcoxon tests. **f** Barplot showing enrichment of the least and most heterogeneous GM12878 100kb windows in PMDs and non-PMDs. Shown are significant p-values from Wilcoxon tests.
(TIF)

**S5 Fig. Periodic DNA methylation patterns are observed in PMDs within single molecules. a** Plot of mean single-read distance-dependent correlation between CpG sites within 500bp of each other for 714, 789 high heterogeneity GM12878 reads and 407, 536 low heterogeneity GM12878 reads. **b** Plot of the power spectral density for the mean distance-dependent correlation associated with the high heterogeneity reads. **c** Magnification of the region indicated in **b** showing the power spectrum peak at 0.0335 (solid line) and the frequency range for which the power spectrum takes at least half of this maximum value (dotted lines). **d** Plot of mean single-read distance-dependent correlation between CpG sites within 2kb of each other for 1, 007, 580 GM24385 reads aligning entirely within PMDs. **e** Magnification of the region indicated in **d** (i.e. mean single-read distance-dependent correlation for distances 1kb-2kb in GM24385 PMD reads).
(TIF)

**S6 Fig. Simple proof-of-principle models suggest that unphasing of nucleosomes between molecules could result in periodic single-molecule methylation patterns being obscured in bulk data. a** Schematic of model of the interaction between DNA methylation and nucleosomes in the case of perfect nucleosomal phasing between molecules (see Methods). **b,c** single-molecule (**b**) and bulk (**c**) recurrence plots derived from the perfect phasing model. Here $\varepsilon = 0.06$ for comparison to recurrence plots associated with real data. **d** Model of completely

unphased nucleosomes (Methods). Here it is assumed that nucleosomes are positioned randomly in each molecule. **e,f** Single-molecule (**e**) and bulk (**f**) recurrence plots derived from the completely unphased model. Again $\varepsilon = 0.06$ for comparison to recurrence plots associated with real data. In **b,c,e,f**, arrows 1 and 2 indicate example regions where the correlations for distances $x$ and $y$ are similar in the single-molecule data generated from the models. Arrow 3 indicates an example region where the correlations for distances $x$ and $y$ are dissimilar in the single-molecule data generated from the models.
(TIF)

## Acknowledgments

We thank Chris Ponting, Marcus Wilson, Andreas Kapourani, Nick Gilbert, Jim Allan and members of the Sproul and Grima lab for helpful discussions about the manuscript.

## Author Contributions

**Conceptualization:** Ramon Grima, Duncan Sproul.

**Formal analysis:** Lyndsay Kerr.

**Methodology:** Lyndsay Kerr.

**Resources:** Ioannis Kafetzopoulos.

**Supervision:** Ramon Grima, Duncan Sproul.

**Visualization:** Lyndsay Kerr.

**Writing – original draft:** Lyndsay Kerr, Ramon Grima, Duncan Sproul.

**Writing – review & editing:** Lyndsay Kerr, Ramon Grima, Duncan Sproul.

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
