## [Decision Letter · Decision Letter 0]

2 Jun 2023

Dear Dr Sproul,

Thank you very much for submitting your Research Article entitled 'Genome-wide single-molecule analysis of long-read DNA methylation reveals heterogeneous patterns at heterochromatin' to PLOS Genetics.

The manuscript was fully evaluated at the editorial level and by independent peer reviewers. The reviewers appreciated the attention to an important topic but identified some concerns that we ask you address in a revised manuscript.

We therefore ask you to modify the manuscript according to the review recommendations. Your revisions should address the specific points made by each reviewer.

Yours sincerely,

Marnie E. Blewitt

Academic Editor

PLOS Genetics

John Greally

Section Editor

PLOS Genetics

Reviewer's Responses to Questions

**Comments to the Authors:**

Reviewer #1: In this manuscript, Kerr et al. use nanopore long-read DNA sequencing of a human lymphoblastoid cell line to investigate the variability of 5mCG along single molecules (mean length 25 kb). Focusing mainly on the correlation between adjacent CpG sites to measure heterogeneity, the authors extend previous results from short read bisulfite sequencing to show that heterochromatin has more heterogeneous methylation than euchromatin. The long reads enable them to show nucleosomal phasing of methylation over several kb. The methods and results are presented in excellent detail and the analysis is well done. While the biological insights are limited and other long-read papers have touched on single-molecule 5mCG heterogeneity and nucleosome phasing (Lee et al. 2020, Altemose et al. 2022, recently Weigert et al. 2023...), it is to my knowledge the first study to quantify these patterns systematically and genome-wide with long reads.

Comments:

1. The coefficient of variation does not seem to be a good metric to quantify heterogeneity within a read. For binary data, the mean and number of CpGs already give an excellent account of the heterogeneity (standard deviation is maximum for a mean of 0.5, and null for means of 0 or 1). Using the coefficient of variation unnecessarily weighs in favour of reads with the lowest methylation. For instance, a read with 99 mCGs and 1 CG (1,1,1…,1,0) would have the same standard deviation (0.1) as a read with 1 mCG and 99 mCGs (0,1,1,...,1), and intuitively it makes sense to consider them as being similarly “heterogeneous”. However the coefficients of variation would be very different: 0.101 for read1 (considered not very variable in the manuscript), and 10 for read2 (considered very variable in the manuscript). The single read correlation calculation should not be affected by this issue, so the rest of the analysis should be fine.

2. Using Nanopolish to call methylation may be problematic for two reasons. 1) Nanopolish looks at k-mers (5 or 6 bases) and cannot tell whether adjacent CpG sites have divergent methylation states, it just calls a probability for the k-mer. This might be particularly problematic when looking at short range correlations (lag-1 correlation, and the nucleosome patterning). 2) Nanopolish sometimes has a lot of low confidence calls. Here they would be filtered out, but it’s possible this introduces bias against certain motifs/genomic contexts/… Using the latest R9 remora models may circumvent these two issues. At least these potential issues with Nanopolish can be discussed and explored.

3. Could you apply a Fourier Transform to the autocorrelation in Fig5 to find the period? Is it possible to detect the 10-bp period one would expect from the DNA helix, as has been reported with short reads? Can you estimate the period in different chromatin contexts, to determine whether nucleosome spacing varies between euchromatin and heterochromatin (and finer ChromHMM distinctions)?

4. typo in Fig5g title

5. some formatting issues in the bibliography (inconsistent capitalisation, ref57, etc)

Reviewer #2: Kerr et al. have presented a really careful and robust analysis of single-molecule of 5mC patterns in data from ONT whole genome sequencing of the GM24385 cell line. The manuscript is very clearly written and presented, with rationale for experimental design and choices being clearly explained throughout. The authors find intra-molecular 5mC heterogeneity, that was not possible to resolve from the analysis of bulk data alone. They find that this heterogeneity is largely enriched in ChromHMM annotated heterochromatin states, and further resolve this observation to show the greatest heterogeneity in previously described partially methylated domains. The authors observe an oscillatory pattern in single molecule 5mC distribution, and hypothesise that this could be the result of differential nucleosome phasing between molecules, and could underlie the observed 5mC heterogeneity. The authors observations are compelling, and I think that the following suggestions would enhance the manuscript:

Can the authors analyse available NanoNOMe data to directly test their hypothesis that nucleosomes are the likely source of the intra-molecular 5mC heterogeneity observed in heterochromatic regions, by analysing 5mC and nucleosome occupancy on single molecules. These data have been generated in a different cellular context, but would be an important analysis nonetheless.

Authors have analysed data from a single transformed cell line, GM24385. As authors have stated in their introduction, there are differences in both 5mC distribution and heterogeneity in cancer. How generalisable is this phenomenon? I would be interested to see whether the observed heterogeneity and oscillatory patterns in heterochromatin could be observed in a primary cells or pluripotent cells if the data is available, particularly because changes in partially methylated domains are hallmarks of cellular differentiation states.

Minor comments:

Authors state read count and mean read length of the data analysed. Could they also add the genome coverage obtained in the data analysed.

When authors were comparing inter- and intra-molecular heterogeneity, they looked at di values for the highest and lowest correlating 10% 100kb windows. I’d like to see what this correlation looks like genome wide, in the more intermediate windows too.

Reviewer #3: In this submission, the authors present a series of computation analyses on Nanopore sequencing data from a single individual that leads them to conclude methylation heterogeneity is common in heterochromatic regions of the genome, and that such heterogeneity appears to follow some periodic pattern that resemble those that nucleosomes appear to cause.

The questions that the authors address are interesting and I am convinced their analyses and the conclusions they reach from them are interesting for the research community, but, in my opinion, the current version of the manuscript still requires work to be ready for publication. I hope that my comments below will help the authors achieve that goal.

Replication.

All analyses are conducted on data from a single individual. Why didn’t the authors use the publicly available Nanopore sequencing data for GM12878, from which the chromHMM-defined regions are obtained?

Differences between bulk and per-read analysis.

The first result presented in the manuscript is that heterogeneity estimates from single-read data are higher than those obtained from bulk values. I am not sure, however, if the conclusion that the authors reach from this observation is warranted. According to them, this indicates that bulk values are "underestimates", but, as far as I understand it, there is no reason to believe that the per-read analysis leads to the actual ground truth value. It is possible that the bulk values correct for the potential methylation call errors that the per-read estimates cannot, and thus they are closer to the real values. Unless some other analysis supports that the difference between the two methods?

Causes for the observations.

The authors put a great deal of effort into attempting to explain the phenomena that they report. I am not sure, however, whether the analyses that they conduct allow them to conclude much regarding causality.

Bulk methylation.

Between lines 280 and 296, the authors address whether the patterns they observe originate from the bulk methylation levels by generating synthetic reads in which the probability of each site to be methylated depended solely on the bulk methylation level at that site. As far as I understand it, based on visual inspection of Figure 5d, the oscillation patterns detected on such synthetic data match very closely those in Figure 5a except perhaps at very close distances. The frequency of the oscillation, in particular, appears to be spot on, and that is, in my opinion, the most relevant observation. In addition, the difference in close distances can be easily explained if one considers that contiguous or close to contiguous sites (such as in CpG islands) are much more likely to be highly linked than randomly, but at distances where that coupling would be less likely the patterns appear to match. I think it is not warranted that the authors claim the patterns "differ". It may well be that bulk methylation alone cannot explain the entirety of the pattern (since there are certainly non-random procedures at work), but that does not mean it is not a cause, even if only partially.

Nucleosomes.

The alternative explanation that the authors attempt to test is whether the patterns could be derived from nucleosome organisation. In fact, that is the explanation that the authors appear to favour, as the conclusions clearly state: "We propose that the nature of this heterogeneity suggests a role for nucleosomes in shaping DNA methylation patterns at the single-molecule level, particularly in heterochromatin". For that reason, it feels strange that all plots leading to this prominent conclusion are placed exclusively on the supplementary information instead of as part of the main figures 5 or 6. If the results are so important that they lead to one of the main conclusions of the manuscript, they should be featured accordingly, which would also help their scrutiny.

Although not explicitly stated in the conclusions, the authors mention at the end of the results section that the "patterns differ from those predicted from mean bulk DNA methylation and could result from differences in nucleosome phasing between molecules" because they consider the recurrence plots generated from the artificial "unphased nucleosome" reads to be "in qualitative agreement" with those from the real reads. Although it is difficult to compare "qualitatively" (and specially since the plots are not presented closely), I would personally say that Figure 5f is more "qualitatively similar" to 5e than S5h is to 5f.

Aside from that, the analysis itself confuses me. I fail to see how this largely qualitative analysis says much about the role of nucleosomes themselves. If I understood it correctly, the analysis involves artificially creating periodic (or pseudo-periodic) patterns of methylation, and see if those somehow match patterns that have already been shown to be periodic. Whether those pseudo-periodic patterns arise from nucleosomes or something else does not seem to really play a role here.

The creation of the "phased" reads also confuses me. If all reads have the "nucleosomes" placed in the exact same positions and the methylation depends exclusively on where those "nucleosomes" fall, are these reads not all exactly the same one? And is it really worth it to check for "bulk" patterns in this case if all reads will match exactly? What is the difference between a single read and multiple that have the exact same methylation state at each position?

Related to the creation of those reads, does the statement in Line 656 that "we assumed that every DNA base on the molecules could be methylated" imply that literally any base, no matter its nucleotide, was considered methylated in the linker sequences between the "nucleosomes"? If that obviously unrealistic assumption is indeed made, I would appreciate it if there was some validation of its implications.

In addition, in the discussion (lines 408 to 411) the authors state that sequence influences nucleosome positioning. If that is the case, would that not imply that nucleosomes would tend to be phased (i.e., located in similar loci on multiple molecules, since those loci would contain similar, if not identical, sequence)?

CpG density.

I feel that the analysis reported here largely fail to account for the effect of CpG density on the methylation state of particular regions. For instance, when comparing heterogeneity values between heterochromatic and euchromatic regions, it would make sense to do so on equivalently dense windows, since CpG islands, for example, will affect the results. Also, the specific heterogeneity metrics, such as the "correlation" or RTS do not take into account how far away the "neighbouring" sites are, and one would expect that contiguous sites are more likely to be in sync than those that are potentially thousands of bases away (on 100 kb windows, a minimum threshold of 100 CpG sites can lead to very distant "neighbours").

**Have all data underlying the figures and results presented in the manuscript been provided?**

Reviewer #1: Yes

Reviewer #2: Yes

Reviewer #3: Yes

PLOS authors have the option to publish the peer review history of their article (what does this mean?). If published, this will include your full peer review and any attached files.

Reviewer #1: **Yes: **Quentin Gouil

Reviewer #2: **Yes: **Natasha Jansz

Reviewer #3: No

---

## [Decision Letter · Decision Letter 1]

4 Sep 2023

Dear Dr Sproul,

We are pleased to inform you that your manuscript entitled "Genome-wide single-molecule analysis of long-read DNA methylation reveals heterogeneous patterns at heterochromatin that reflect nucleosome organisation" has been editorially accepted for publication in PLOS Genetics. Congratulations!

Yours sincerely,

Marnie E. Blewitt

Academic Editor

PLOS Genetics

John Greally

Section Editor

PLOS Genetics

Comments from the reviewers (if applicable):

Reviewer's Responses to Questions

**Comments to the Authors:**

Reviewer #2: I thank the authors for their thorough response to my queries; they have answered all of my questions with clarifications and additional data. I do think the discussion points added in review have significantly softened the statements around causality and mechanism, and now, rather, raise interesting questions to be addressed in future studies. I believe the manuscript is significantly improved.

Reviewer #3: The authors have done a good job in addressing my comments. I have no further comments to make.

**Have all data underlying the figures and results presented in the manuscript been provided?**

Reviewer #2: Yes

Reviewer #3: Yes

PLOS authors have the option to publish the peer review history of their article (what does this mean?). If published, this will include your full peer review and any attached files.

Reviewer #2: **Yes: **Natasha Jansz

Reviewer #3: No

**Data Deposition**

http://datadryad.org/submit?journalID=pgenetics&manu=PGENETICS-D-23-00502R1

**Press Queries**

---

## [Editor Report · Acceptance letter]

27 Sep 2023

PGENETICS-D-23-00502R1 

Genome-wide single-molecule analysis of long-read DNA methylation reveals heterogeneous patterns at heterochromatin that reflect nucleosome organisation 

Dear Dr Sproul, 

We are pleased to inform you that your manuscript entitled "Genome-wide single-molecule analysis of long-read DNA methylation reveals heterogeneous patterns at heterochromatin that reflect nucleosome organisation" has been formally accepted for publication in PLOS Genetics! Your manuscript is now with our production department and you will be notified of the publication date in due course.

With kind regards,

Timea Kemeri-Szekernyes

PLOS Genetics

On behalf of:
